# Assessing the conservation and targets of putative sRNAs in *Streptococcus pneumoniae*

Matthew C. Eichelman,[1] Michelle M. Meyer[1]

**ABSTRACT** RNA regulators are often found in regulatory networks and mediate growth and virulence in bacteria. Small RNAs (sRNAs) are non-coding RNAs that modulate translation initiation and mRNA degradation by base pairing. To better understand the role of sRNAs in pathogenicity, several studies identified sRNAs in *Streptococcus pneumoniae*; however, little functional characterization has followed. This study's goals are to (i) survey putative sRNAs in *S. pneumoniae*; (ii) assess the conservation of these sRNAs; and (iii) examine their predicted targets. Three previous studies in *S. pneumoniae* identified 287 putative sRNAs by high-throughput sequencing. This study narrows the candidates down to 58 putative sRNAs. BLAST analysis indicates that the 58 sequences are highly conserved across the *S. pneumoniae* pangenome, and 25 are identified sporadically in other Streptococcus species. However, only two have corresponding sequences identified across several Streptococcus species. We used four RNA-target prediction programs to predict targets for each of the 58 putative sRNAs. Across all probable predictions, six sRNAs have overlapping targets predicted by multiple programs, four targeting numerous transposase-encoding transcripts. sRNAs targeting transposase-encoding transcripts display nearly identical and perfect base pairing. One sRNA, M63 (Spd_sr37), has several probable targets in the CcpA regulon, a network responsible for global catabolite repression, suggesting a possible biological function in carbon metabolism control. Each M63-target interaction exhibits unique base pairing, increasing confidence in the biological relevance of the result. This study produces a list of *S. pneumoniae* putative sRNAs whose predicted targets suggest functional significance in transposon and carbon metabolism regulation.

**IMPORTANCE** Previous studies identified many small RNA candidates in *Streptococcus pneumoniae*, several of which were hypothesized to play a role in *S. pneumoniae* virulence. Due to the differing sequencing methods, diverse inclusion criteria, *S. pneumoniae* strain differences, as well as limited follow-up, it is unclear to what extent candidates identified in different studies have overlapping sequences and functions, and their biological relevance remains ambiguous. This research aims to consolidate the candidate sRNAs across these studies and focuses attention on those that are likely to be regulatory and associated with virulence. This study's findings enhance our knowledge of the conservation of small regulatory RNAs across the many *Streptococcus pneumoniae* strains and highlight a handful that appear likely to have a role in growth or virulence.

**KEYWORDS** sRNA, *Streptococcus*, transposon

*S*treptococcus pneumoniae is a Gram-positive bacterium that causes various diseases including pneumonia, meningitis, bacteremia, otitis media, and sinusitis. Invasive pneumococcal disease is particularly dangerous in children and the elderly (1), and in 2004, it was responsible for approximately 4 million illness episodes, 445,000 hospitalizations, and 22,000 deaths in the United States (2). In 2016, *S. pneumoniae* was the leading

Address correspondence to Michelle M. Meyer, m.meyer@bc.edu.

The authors declare no conflict of interest.

See the funding table on p. 14.

cause of lower respiratory infection morbidity and mortality globally, causing over a million deaths (3). Despite the threat *S. pneumoniae* poses, important components of regulation relating to metabolism and virulence remain less well characterized. Small regulatory RNAs (sRNA) are sequences of 40–500 nucleotides (nt) in length (4) that can be transcribed by 5′-UTRs, 3′-UTRs, coding, and non-coding sequences (5). However, studies seeking to identify sRNAs tend to focus on intergenic regions because RNAs transcribed in regions lacking an ORF are assumed to be more likely to be functional regulators, whereas those identified within ORFs may be an intermediate RNA decay product from a protein-encoding transcript. Among the different types of sRNAs are *trans*-encoded and *cis*-encoded RNAs. *Trans*-encoded sRNAs regulate genes from distant regions, often with imperfect complementarity, allowing them to interact with more than one target (6). *Cis*-encoded sRNAs act on the mRNA transcript encoded by the opposite DNA strand, leading to perfect complementarity (7) (Fig. 1A). sRNAs modulate the expression of target mRNAs by base pairing to sequester a ribosome-binding site or accelerate decay (8). Some sRNAs are dependent on a chaperone like the Hfq or FinO family proteins that have a well-characterized role in aiding the formation of duplexes between sRNAs and their mRNA targets in Enterobacteriaceae. However, the role of RNA chaperones in Gram-positive bacteria is substantially less clear. Such proteins are frequently not present, and when they are present, their functionality is often substantially different from that observed in Gram-negative organisms (9). In *S. pneumoniae*,

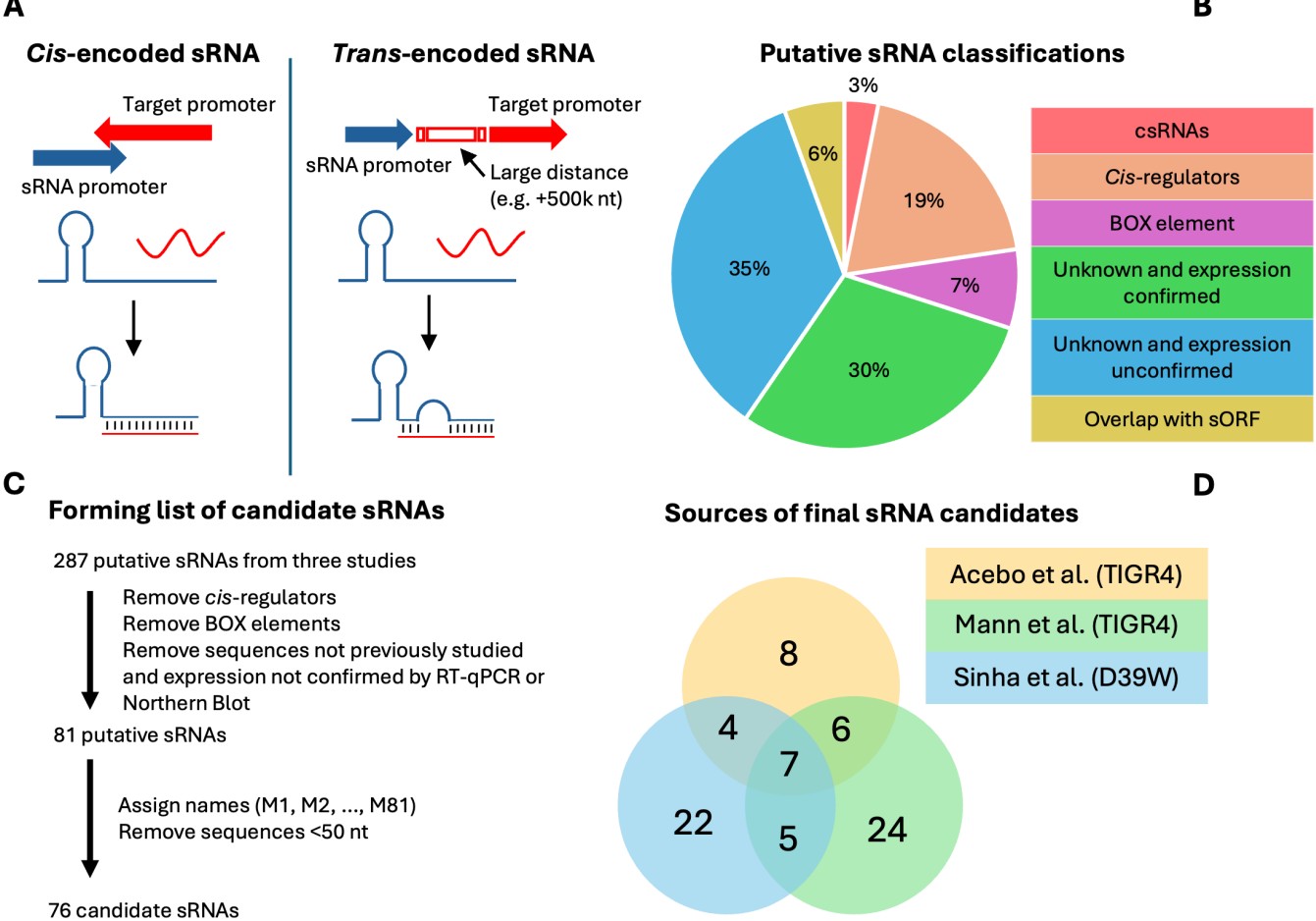

FIG 1 (A) The *cis*-encoded sRNA, found on the strand opposite to the mRNA coding strand, binds the target mRNA with perfect complementarity. The *trans*-encoded sRNA, expressed from a region distant to the target, binds with imperfect complementarity. (B) The classifications of the 287 putative sRNAs. (C) Process of narrowing the 287 putative sRNAs to a list of 76 candidates for further analysis. (D) Among the candidates, 7 were identified by all three studies and 15 by two of the three studies.

recent work shows KH domain proteins, like KhpA and KhpB, are associated with sRNAs (10), and sRNAs have been associated with the exonuclease Cbf1 (11). However, there are no confirmed sRNA chaperone homologs in the *S. pneumoniae* genome (12).

Identification of putative sRNAs in *S. pneumoniae* has been performed several times, but follow-up characterization has been limited. The exception is a group of sRNAs called cia-dependent sRNAs (csRNAs) that have been studied across Streptococcus, with considerable work done in *S. pneumoniae*. The csRNAs are controlled by the CiaRH two-component system (TCS) that is involved in natural competence and general virulence (13–15). In *S. pneumoniae,* the CiaRH TCS expresses five sRNAs, with experimentally verified targets, that prevent autolysis triggered by various conditions, like the presence of deoxycholate, to allow the maintenance of the stationary growth phase (16). The csRNAs have also recently been implicated in promoting Zn homeostasis (17). Three previous studies identified hundreds of additional putative sRNAs using diverse inclusion criteria. Some of these sequences may be annotated as homologs of RNA families such as Pyr elements (RF00515) or TPP riboswitches (RF00059), BOX elements (AT-rich repeats that are highly transcribed), or ribosomal protein leaders (sequences in the 5′-UTR of ribosomal protein transcripts that control the concentration of the ribosomal protein) (18–20). However, it remains unclear which of the remaining sequences are regulatory. Here we find that 58 of the putative sRNAs are highly conserved in the *S. pneumoniae* pangenome, and we predict the mRNA targets of these sRNAs. The predictions suggest four putative sRNAs are likely to interact with mRNAs coding for transposases. An additional three putative sRNAs have likely targets that include transcripts encoding an $H^+/Cl^-$ exchange transporter, RuvB (involved in DNA recombination), and genes involved in carbon metabolism regulation (CcpA regulon), respectively.

## RESULTS AND DISCUSSION

### *S. pneumoniae* genome contains +70 putative sRNAs

In assessing which previously identified sRNA candidates are likely to have a biological function and prioritizing candidates for further investigation, we examined a pool of 287 putative sRNAs originating from three studies (18, 21, 22) (Additional Data File S1). We note that 65% are functionally uncharacterized, whereas the other 35% may be annotated as homologs of *cis*-regulators, BOX elements, csRNAs, or overlap with small open reading frames (sORF) (Fig. 1B). We further narrowed this pool to a list where each sRNA has at least one of three attributes: (i) sequence identified in multiple studies, (ii) expression confirmed by Northern blot or RT-qPCR, or (iii) sequence characterized as a csRNA or overlapping with a sORF (23). *Cis*-regulators and BOX elements were also excluded from our list. Thus, the new list contains 81 putative sRNAs that were assigned names "M1" through "M81". However, five sequences are <50 nt and were subsequently removed from the list, leaving a final total of 76 candidate sRNAs (Fig. 1C). We observe that many of these sRNAs were found in a single study emphasizing the different sequencing strategies and inclusion criteria of the previous studies (Fig. 1D). After compiling the final list for further analysis, we conclude there are over 70 putative sRNAs in the *S. pneumoniae* genome (Additional Data File S2).

### Majority of sRNA candidates are conserved across the *S. pneumoniae* pangenome

To increase our confidence in the biological relevance of the putative sRNAs and prioritize them for further investigation, we assessed the conservation of the candidate sRNAs across the genomes of 385 *S. pneumoniae* strains. BLAST (24) analysis indicated 70/76 candidates are present in the genomes for a majority of 385 *S. pneumoniae* strains (Fig. 2A) (25, 26). Among these 70, only 60 candidates appear to be non-repetitive sequences, all of which display average sequence identity >97% to the best hit in each genome, indicating the sequences are highly conserved across the *S. pneumoniae* pangenome (Table 1). Interestingly, all of the 6 sRNAs found in <12 strains were identified

**A**

## 76 sRNA candidates

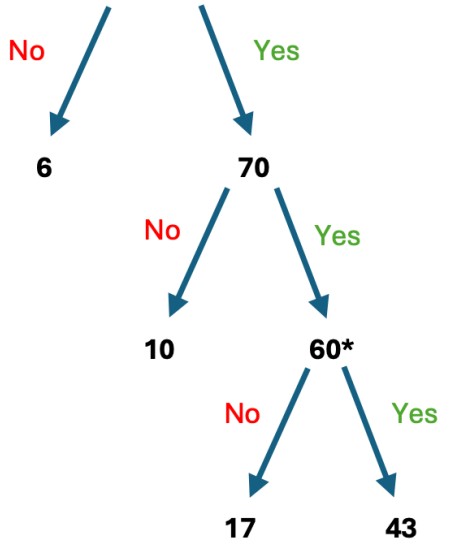

1. <u>Conserved across pangenome</u>: Found in >338 strains?

2. <u>Non-repetitive sequence</u>: ≤15 alignments per genome?

3. <u>Synteny preserved</u>: Are the sequence identities of 1000 nt up and downstream of the sRNA ≥75%

**B**

## Conservation of sRNA candidates

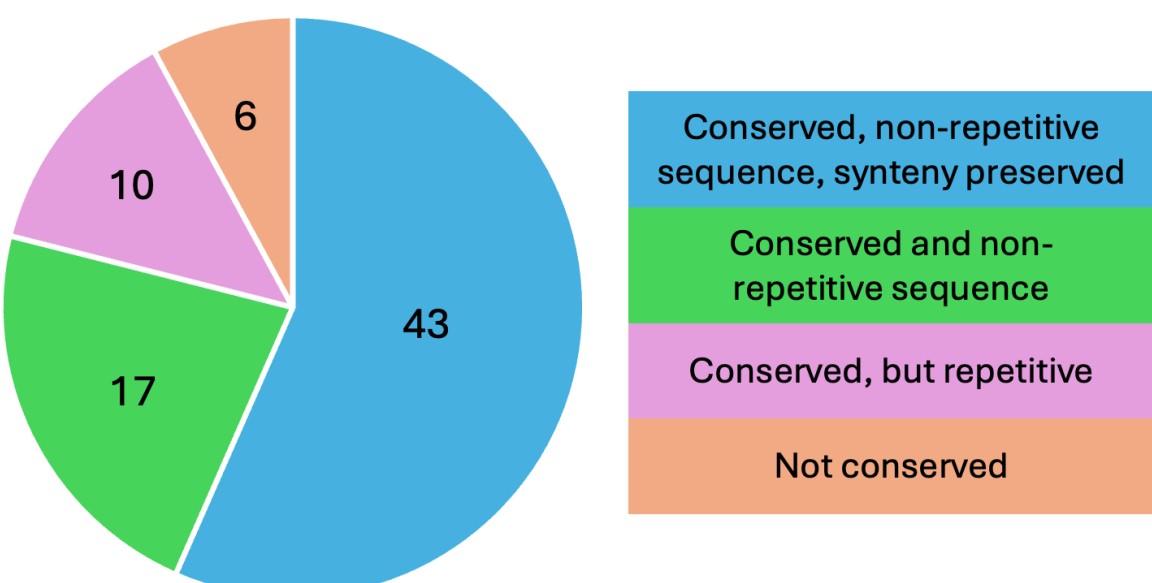

FIG 2 (A) The conservation of sRNA candidates across the *S. pneumoniae* pangenome. The asterisk (*) next to "60" highlights that 2 sRNAs were later removed to form the final list of 58 sRNAs (see Table 1 legend). (B) The four degrees of conservation of the sRNA candidates. "Conserved," "non-repetitive sequence," and "synteny preserved" refer to the criteria for each level of the tree diagram in part A.

in the *S. pneumoniae* strain TIGR4 (Fig. 2), highlighting the possibility for strain-specific sRNAs in *S. pneumoniae*. Two of the final candidates, M8 and M12, were identified as *cis*-regulatory elements later during our analysis and removed, leaving a total of 58 sRNAs for further analysis. From this final group of 58 candidates, we observed that synteny is preserved (see Materials and Methods) across the *S. pneumoniae* strains in 43 of the 58 final candidates. To determine whether any of the sRNAs are conserved in species other than *S. pneumoniae*, we also analyzed a range of other species in

**TABLE 1** The 76 candidate sRNAs and their conservation[a,c]

| In-house ID | Other IDs | TIGR4 coordinates | D39W coordinates | Sequence identity (%) | Genomes with match | Matches per genome | Overlap with sORF |
|---|---|---|---|---|---|---|---|
| M1 | srn061/F7/CcnE | 209768–209916 | 212278–212426 | 99.99 | 385[1] | 1 | |
| M2 | srn135 | 438151–438275 | | 100.00 | 8 | 1 | |
| M3 | srn151 | 501260–501363 | | 96.32 | 385 | 19 | |
| M4 | F15 | 501732–501843 | | 99.04 | 385[1] | 1 | |
| M5 | srn206 | 781187–781304 | | 99.77 | 384[1] | 1 | |
| M6 | srn231 | 853483–853605 | | 97.97 | 348[1] | 1 | |
| M7 | srn235/F59/Spd_sr49 | 869478–869804 | 825803–826129 | 99.39 | 385[1] | 1 | |
| M8[b] | srn241/F35 | 909029–909179 | | 98.93 | 385[1] | 1 | |
| M9 | srn254/F38/Spd_sr17 | 956782–956927 | 912572–912717 | 97.90 | 385[1] | 1 | rio56 |
| M10 | srn266/Spd_sr55 | 1035442–1035581 | 1697549–1697677 | 97.33 | 385 | 15 | |
| M11 | srn277/F41 | 1071102–1071213 | | 91.09 | 4 | 1 | |
| M12[b] | srn351/R21/Spd_sr78 | 1461066–1461188 | 1404039–1404161 | 99.93 | 385[1] | 1 | |
| M13 | srn477/Spd_sr105 | 1984829–1984968 | 1873277–1873416 | 99.86 | 385[1] | 1 | |
| M14 | trn0978 | 1986229–1986310 | | 99.70 | 385[1] | 1 | |
| M16 | srn491/Spd_sr106 | 2005533–2005682 | 1892401–1892550 | 99.86 | 385[1] | 1 | |
| M17 | srn502/F66/Spd_sr109 | 2086085–2086325 | 1972860–1973100 | 99.55 | 385 | 2 | |
| M18 | srn503/F67/Spd_sr111 | 2086380–2086628 | 1973155–1973403 | 98.41 | 385 | 2 | |
| M19 | trn0012/CcnC | 24164–24262 | 23967–24065 | 99.62 | 385 | 2 | |
| M20 | trn0057 | 124989–125050 | | 97.19 | 385 | 1 | |
| M21 | Spd_sr18 | | 134413–134576 | 96.75 | 385 | 92 | |
| M22 | trn0157/F10 | 284239–284316 | | 99.67 | 374[1] | 1 | |
| M23 | trn0332/F25/Spd_sr42 | 623244–623344 | 587440–587542 | 95.89 | 378[1] | 1 | |
| M24 | trn0485/F60 | 950115–950195 | | 99.99 | 385[1] | 1 | |
| M25 | trn0696 | 1423633–1423713 | | 93.05 | 364 | 1 | |
| M27 | trn0830/R12 | 1731041–1731439 | | 94.32 | 385 | 209 | |
| M28 | trn0935/R9 | 1903548–1903637 | | 99.82 | 385 | 19 | |
| M29 | trn1025 | 2048577–2048643 | | 99.99 | 366[1] | 1 | |
| M30 | R7 | 1791010–1791079 | | 98.58 | 385 | 6 | |
| M31 | R8 | 1892645–1892716 | | 99.54 | 9 | 1 | |
| M32 | R14 | 1034021–1034099 | | 98.69 | 385[1] | 1 | |
| M33 | R17/Spd_sr69 | 1277241–1277387 | 1217391–1217537 | 97.32 | 385 | 79 | |
| M34 | R18/Spd_sr72 | 1364575–1364763 | 1299979–1300167 | 98.30 | 379[1] | 1 | |
| M36 | F1 | 91593–91663 | | 99.09 | 361 | 1 | |
| M37 | F3 | 117143–117247 | | 98.43 | 385 | 1 | |
| M39 | F6 | 130439–130494 | | 98.30 | 383[1] | 1 | |
| M40 | F8/CcnA | 228626–228807 | 231143–231324 | 97.12 | 385[1] | 2 | |
| M41 | F11 | 286614–286707 | | 99.13 | 384[1] | 1 | |
| M42 | F14 | 499570–499688 | | 99.77 | 339[1] | 1 | |
| M43 | F18 | 538437–538491 | | 97.84 | 385 | 7 | |
| M44 | F19/Spd_sr33 | 543000–543149 | 508238–508388 | 99.11 | 385[1] | 1 | rio39 |
| M45 | F22 | 592573–592711 | | 99.44 | 385[1] | 1 | |
| M46 | F24 | 610528–610659 | | 98.04 | 385 | 12 | |
| M47 | F31 | 810811–810861 | | 98.29 | 385 | 12 | |
| M48 | F33 | 863736–863817 | | 98.28 | 385[1] | 1 | |
| M49 | F36 | 941435–941486 | | 99.66 | 385[1] | 1 | |
| M50 | F39 | 972498–972606 | | 98.15 | 385[1] | 1 | |
| M51 | F40 | 1063101–1063150 | | 100.00 | 4 | 1 | |
| M52 | F43/Spd_sr63 | 1216148–1216245 | 1170289–1170386 | 99.87 | 385[1] | 1 | |
| M53 | F45 | 1408204–1408274 | | 100.00 | 384[1] | 1 | |
| M54 | F48 | 1778293–1778426 | | 99.27 | 385 | 36 | |
| M56 | F53 | 588512–588589 | | 99.92 | 385 | 26 | |

*(Continued on next page)*

**TABLE 1** The 76 candidate sRNAs and their conservation[a,c] (*Continued*)

| In-house ID | Other IDs | TIGR4 coordinates | D39W coordinates | Sequence identity (%) | Genomes with match | Matches per genome | Overlap with sORF |
|---|---|---|---|---|---|---|---|
| M57 | F55 | 1696066–1696161 | | 97.13 | 385 | 48 | |
| M58 | F56 | 158993–159089 | | 98.09 | 11 | 1 | |
| M59 | F61 | 972327–972383 | | 98.82 | 385[1] | 1 | |
| M60 | F62 | 995726–995786 | | 100.00 | 9 | 1 | |
| M61 | Spd_sr5 | | 39980–40081 | 97.40 | 358[1] | 1 | rio3 |
| M62 | Spd_sr6 | | 41494–41558 | 97.76 | 384 | 7 | |
| M63 | Spd_sr37 | | 131773–131841 | 99.83 | 385[1] | 1 | rio16 |
| M64 | Spd_sr14 | | 149223–149340 | 99.59 | 385[1] | 1 | rio17 |
| M65 | CcnB | | 231331–231426 | 99.71 | 385 | 3 | |
| M66 | Spd_sr24 | | 231853–232034 | 99.34 | 371 | 1 | |
| M67 | CcnD | | 233715–233808 | 99.89 | 385[1] | 1 | |
| M68 | Spd_sr31 | | 476085–476234 | 98.76 | 385[1] | 1 | |
| M69 | Spd_sr47 | | 825484–825544 | 99.38 | 385[1] | 2 | |
| M70 | Spd_sr60 | | 1079136–1079199 | 99.99 | 383[1] | 1 | |
| M71 | Spd_sr67 | | 1212230–1212526 | 98.77 | 365[1] | 2 | |
| M72 | Spd_sr71 | | 1264469–1264569 | 99.06 | 385 | 37 | |
| M73 | Spd_sr81 | | 1464371–1464684 | 99.17 | 385[1] | 1 | |
| M74 | Spd_sr83 | | 1528062–1528186 | 99.64 | 385[1] | 1 | rio82 |
| M75 | Spd_sr84 | | 1595446–1595563 | 98.99 | 385[1] | 1 | |
| M76 | Spd_sr89 | | 1673201–1673322 | 99.92 | 385[1] | 1 | rio86 |
| M77 | Spd_sr96 | | 1759320–1759411 | 100.00 | 385[1] | 1 | |
| M78 | Spd_sr108 | | 1913212–1913442 | 97.60 | 385 | 52 | |
| M79 | Spd_sr110 | | 1973001–1973113 | 99.16 | 385 | 2 | |
| M80 | Spd_sr112 | | 1973343–1973456 | 98.78 | 385 | 2 | |
| M81 | Spd_sr116 | | 2020113–2020228 | 99.98 | 385[1] | 1 | |

[a]The TIGR4 coordinates indicate the sequence location in the NC_003028.3 genome. Likewise, D39W coordinates correspond to the NC_008533.2 genome. The "Sequence identity" and "Matches per genome" are averages across the 385 strains. Preservation of synteny indicated by [1] in the "Genomes with match" column.

[b]M8 was later identified as a *cis*-regulator (pyrR element: RFAM:RF00515), and M12 was identified as a $Mn^{2+}$ responsive riboswitch (28) (RFAM:RF000080). The "Overlap with sORF" column indicates the sRNA overlaps with a previously identified sORF (23). White rows correspond to the 58 final sRNAs that are conserved in many *S. pneumoniae* strains, but have fewer than 15 instances per genome.

[c]In the "Other IDs" column, sRNAs with the prefix "srn," "trn," "R," and "F" were identified in TIGR4 (18, 21), and sRNAs with the prefix "Spd_sr" and "Ccn" were identified in D39W (22).

the *Streptococcus* genus. Only two of the candidate sRNAs (M77 and M81) align with sequence identities > 65% to each of *S. pyogenes*, *S. mutans*, *S. suis*, *S. mitis*, *S. oralis*, and *S. gordonii,* and 25 additional sRNA candidates can be identified in a subset of these organisms (Table S1). In addition, we see a higher number of sRNAs conserved in the more closely related species, like *S. oralis* and *S. mitis*, than the more distant species like *S. mutans* and *S. pyogenes*. Thus, it appears over half of the sRNAs are unique to *S. pneumoniae*, consistent with the narrow distribution of many sRNAs across other bacterial species (27).

## Several sRNA candidates overlap with reported sORFs

Of the 58 final sRNA candidates, 6 of them overlap with previously identified sORFs (Table 1) (23), and one (M64) is complementary to an sORF. Moreover, we see that 5 of the 7 sORFs are found in nearly all 385 strains with 1 alignment per genome and a high degree of conservation with nucleotide and peptide sequence identities > 98%. The first exception, rio56, is a six-amino acid peptide sequence and is too short for BLAST to produce alignments with an e-value <20, and the second, rio86, has an average peptide sequence identity of 93.6%. Of particular interest is M61 (Spd_sr5/srf-02), which overlaps with rio3, whose expression was shown to promote *in vivo* fitness (23). However, dual-function RNAs, sequences with coding and non-coding functions, are known to exist in other bacteria like SgrT/SgrS and AzuC/AzuR in *Escherichia coli* (29) and Pel RNA

in *Streptococcus pyogenes* (30). Thus, we decided to retain sORF overlapping sRNAs for target analysis.

## RNA target prediction programs struggle to correctly predict validated targets

Several RNA-RNA interaction prediction (RIP) programs have been developed to predict mRNA:sRNA interactions, with newer models displaying the highest accuracies. These include IntaRNA (31–34), CopraRNA (32, 34, 35), sRNARFTarget (36), and TargetRNA3 (37). The programs take various approaches, with newer RIP programs implementing machine learning algorithms. Despite the improvement over time, all the tools have a high false-positive rate (37). Moreover, most of the data on which the models are validated and trained are from Hfq-dependent sRNA networks in *E. coli* that may not be reflective of sRNA-target interactions in organisms without Hfq like *S. pneumoniae*. This poses a challenge to determining the validity of a predicted target through computational methods alone. By examining the targets of multiple programs with different approaches, we hope to increase confidence in the validity of predicted sRNA-target pairs.

As a baseline evaluation of the RIP programs, we compared the known and predicted targets of the csRNAs (within six different *S. pneumoniae* strains, including D39W and TIGR4, where the original sequencing was conducted, see Materials and Methods) using IntaRNA, sRNARFTarget, and TargetRNA3 (38). None of the programs correctly predicted any of the known csRNA targets (SP_2237/SP_RS11435, SP_0090/SP_RS00460, SP_0161/SP_RS00830, SP_0626/SP_RS03070, and SP_1215/SP_RS05965) as the most likely target. If we include the top five most likely targets, then IntaRNA correctly predicts that csRNAs 2 and 3 target SP_RS00460, and both IntaRNA and TargetRNA3 correctly predicted that csRNA4 targets SP_RS00460. We also confirmed that the sequences that we examined are consistent with reported 5′ and 3′ RNA-seq data in TIGR4 (39, 40) to ensure that our inputs were not causing the low accuracy. These results support the existing evidence demonstrating that even the best RIP programs suffer from high false-positive and negative rates but can provide informative results.

## RIP programs predict thousands of sRNA-target pairs

We used multiple programs to make target predictions for the candidates. For all 58 sRNAs, we used IntaRNA, sRNARFTarget, and TargetRNA3 and the in-house sequences generated from the previous sequencing studies (see Materials and Methods). Targets were predicted in six different *S. pneumoniae* strains: TIGR4, D39, and four arbitrarily selected strains from PRJNA514780 (26) (see Materials and Methods). The sRNA sequences used for target prediction are derived from the original strain sequenced (TIGR4 and D39W); thus, small differences between these genomes and the other *S. pneumoniae* strains could affect target prediction in other genomes. However, the amount of sequence variation is very small in these loci across the strains, and the consideration of many features in target prediction minimizes concerns that single-nucleotide changes will drastically affect the predictions across a set of genomes. We also used CopraRNA, but only for the three sRNAs with sequence identities > 65% in at least four related Streptococcus species due to the algorithm's comparative approach to target identification. CopraRNA was used to make predictions for *S. pneumoniae*, *S. pyogenes*, *S. mutans*, *S. suis*, *S. mitis*, *S. oralis*, and *S. gordonii*. Each program produces a variable number of outputs per sRNA per strain/species. IntaRNA and CopraRNA made five predictions (a customizable parameter), sRNARFTarget predicted a probability for every gene in the *S. pneumoniae* transcriptome (>2,000 genes), and TargetRNA3 reported a variable number of targets with a probability and *P*-value above a customizable threshold.

In total, we obtained thousands of predictions, the majority of which have low probabilities (≤0.5) (See Additional Data File S3 to S7). To focus our attention on likely sRNA-target pairs without excluding too many predictions, we settled on targets with

a predicted probability ≥0.7, referred to as probable going forward. We also define the term MPT, most probable target, as the prediction given the highest probability across all predictions for a given sRNA. Lastly, we define a consensus target as a gene that was predicted to be the MPT for an sRNA in at least four of the six *S. pneumoniae* strains. This term only pertains to the predictions made by IntaRNA, sRNARFTarget, and TargetRNA3. We observed that none of the sRNARFTarget predictions are probable. This, in combination with our baseline evaluation of the csRNAs, led us to focus on the predictions made by IntaRNA, TargetRNA3, and CopraRNA when applicable.

In many cases, sRNA pairing can impact the internal base pairing of the mRNA to enable gene expression changes (41). Thus, we assessed whether the putative mRNA target regions are likely to have internal base pairing that may be impacted by sRNA interaction. We used RNAfold (42) to predict the structure of mRNA target sequences, including 25 nucleotides up/downstream of the binding region in the absence of any sRNA partner. We consider a structured region to be a segment of the mRNA sequence displaying internal base pairing (>2 consecutive bases). Across the MPTs predicted by IntaRNA and TargetRNA3, 51 out of 58 sRNAs base pair with a region considered structured, suggesting that the putative sRNA-mRNA interaction may induce structural changes in the secondary or tertiary structures to enable regulation (43).

## sRNAs may play a role in pathogenesis via their targets

Previous studies used transposon insertion mutants to conclude that specific sRNAs may support virulence in TIGR4 (18). We compared our predicted targets for these sRNAs to evaluate these hypotheses. Previous work suggested that eight putative sRNAs (Table 2) play a definitive role in pathogenesis, and some individual target loci were identified by microarray analysis of attenuated sRNA mutants (18). Only three of these sRNAs met our criteria for further investigation (Fig. 2A; Table 2). The others overlap with known *cis*-regulatory elements (F20 and F44), transfer-messenger RNA (F32), or were removed following conservation analysis (F41 and F48). F41 is one of the sRNAs found in <12 strains, and F48 was deemed a repetitive sequence (average of 36 copies per genome). Among the three remaining candidates, one (M1/F7) is a csRNA with an established role in pathogenesis (17, 38, 44). The other two, M45 and M23, are not characterized. M45 is predicted to target the mRNA encoding type IV teichoic acid flippase TacF that is responsible for transporting choline across the cytoplasmic membrane, a nutritional requirement of *S. pneumoniae* (45). M23 was originally reported to target SP_RS08340-50, a putative carbohydrate transporter, based on microarray analysis, but we predict that it targets a transposase-encoding transcript (SP_RS13320) (Table 2). Without validating the targets, the role of M45 and M23 in pathogenesis is unclear; however, the predicted target (*tacF*) of M45 is suggestive of such a role.

To further assess whether specific sRNAs are potentially regulating multiple targets in a previously recognized regulatory response (Table 3), we investigated whether the predicted targets belong to established operons or regulons in TIGR4. The regulons that

**TABLE 2**   A comparison of the in-house target predictions and the putative targets identified by microarray analysis[a]

| Mann study ID | In-house ID | Mann putative target | In-house consensus target |
|---|---|---|---|
| F7 | M1, csRNA5 | | SP_RS06250 |
| F20 | T-box leader | | |
| F22 | M45 | | SP_RS06235 |
| F25 | M23 | SP_RS08340-50 | SP_RS13320 |
| F32 | tmRNA | | |
| F41 | | SP_RS08340-50 | |
| F44 | PyrR-binding site | SP_RS08340-50 | |
| F48 | | | |

[a]The previously identified putative targets SP_RS08340-50 are three neighboring loci involved in carbohydrate transport (carbohydrate ABC transporter permease) and proposed to be collectively regulated by three of the sRNAs (18).

**TABLE 3** The regulons, according to the RegPrecise database, in which targets of M63 are involved[a]

| sRNA | Target gene | Regulon | Position in operon | Target annotation |
|---|---|---|---|---|
| M63 | *adhE*/SP_2026/SP_RS10245 | CcpA | 1 of 1 | Bifunctional acetaldehyde-CoA/alcohol dehydrogenase |
| M63 | *adhE*/SP_2026/SP_RS10245 | Rex | 1 of 1 | Bifunctional acetaldehyde-CoA/alcohol dehydrogenase |
| M63 | *hexA*/SP_0498/SP_RS02450 | CcpA | 1 of 1 | Bacterial Ig-like domain-containing protein |
| M63 | *hexB*/SP_0057/SP_RS00325 | CcpA | 1 of 1 | LPXTG-anchored beta-N-acetylhexosaminidase StrH |
| M63 | *oppA2*/SP_1891/SP_RS09395 | CodY | 1 of 5 | Peptide ABC transporter substrate-binding protein |
| M63 | *glgB*/SP_1121/SP_RS05550 | CcpA | 1 of 4 | 1,4-alpha-glucan branching protein GlgB |
| M63 | *ileS*/SP_1658/SP_RS08185 | T-box(Ile) | 1 of 1 | Isoleucine--tRNA ligase |
| M63 | *carB*/SP_1275/SP_RS06250 | PyrR | 1 of 1 | Carbamoyl-phosphate synthase large subunit |

[a]Predictions made by TargetRNA3 in *S. pneumoniae* strain TIGR4.

appeared the most often are PyrR, CodY, and CcpA, and we noticed that the targets belonging to established operons are always the first or last gene in the operon, with the first gene being more common. This suggests that the sRNAs may be inhibiting translation, typically blocking the ribosome-binding site (e.g., start of an operon), or stabilizing the transcript by binding to the 3′ end of the mRNA, depending on the relative location of interaction (8). Most notably, one sRNA, M63 (Fig. 3A), is predicted to target four different genes in the CcpA regulon (Table 3), genes regulated by the catabolite control protein A (CcpA), an essential transcription factor in Gram-positive bacteria that is responsible for mediating carbon catabolite repression and activation. In *S. pneumoniae*, mutations in CcpA reduce virulence in mouse models (46, 47). We also see that M63 interacts with the different targets in various regions of the sRNA with unique base pairing (Fig. 3B). Lastly, M63 is unique in that it is the only sRNA candidate where every reported target has a probability ≥0.7. Furthermore, this is true for all 6 *S. pneumoniae* strains assessed. M63 also overlaps with the rio16 sORF, indicating that part of this sRNA is translated. The sORF has a conserved sequence with an average sequence identity of 99.7% across the pangenome. However, it remains possible that M63 is a dual-function RNA, both encoding a small protein and regulating the members of the CcpA regulon. The observation that M63 has multiple probable targets acting on a regulon associated with virulence (47) makes this sRNA a high-priority candidate for further investigation.

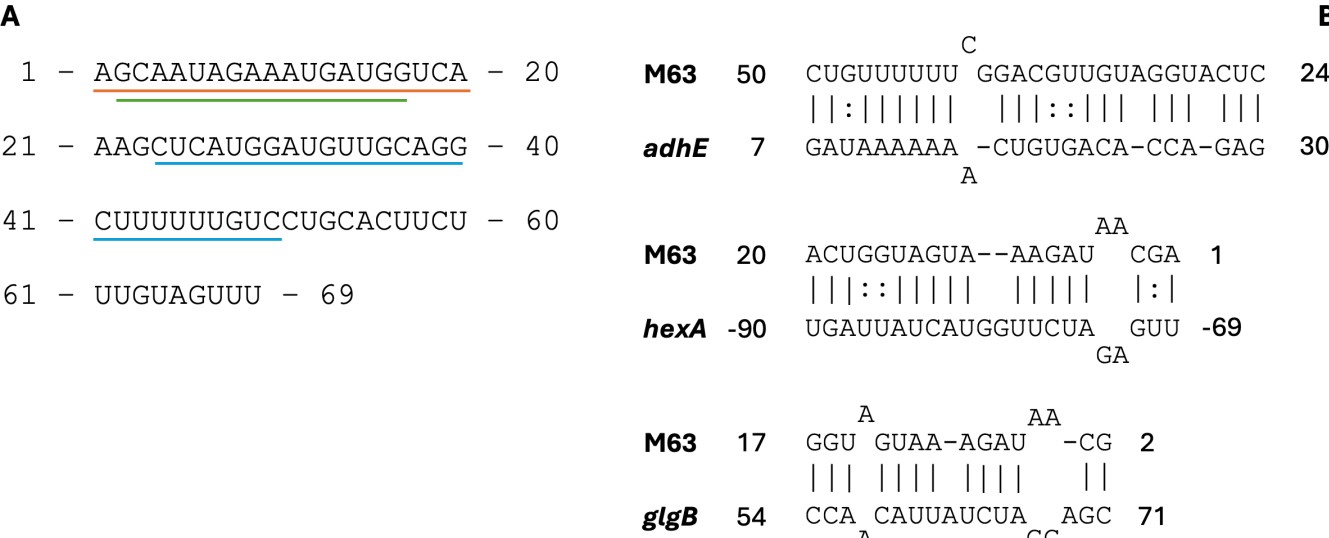

**FIG 3** (A) The M63 sequence with underlined subsequences indicating the regions interacting in part B. Blue corresponds to the *adhE* interaction, orange to *hexA*, and green to *glgB*. (B) Three of the M63 interactions predicted by TargetRNA3 in TIGR4. The gene names are predicted targets of M63 in the CcpA regulon. The numbers on either end are relative positions of the interacting sequences within the full RNA sequence for the sRNA. For the mRNAs, numbering is relative to the mRNA start codon. A ":" indicates a G-U pair.

## Transposase-associated sRNAs are frequent

Among the TargetRNA3 predictions, we noticed a large number of transposase-associated targets. Candidates targeting transposase-encoding transcripts include M10, M47, M62, and M69. These sRNAs are all encoded antisense to an annotated transposase (IL3 or IL30 family), overlapping with the 5′-UTR or first few amino acids of the gene. A subset of these, M10, M47, and M62, shows substantial sequence identity to each other, with M10 having a 3′-extension compared to M47 and M62 (Fig. 4A). Notably, M10, M47, and M62 all have a large number of BLAST hits in the genome, but these sequences did not exceed our threshold of >15 hits in the genome to be considered repetitive sequences. M47 and M62 have consensus targets with both IntaRNA and TargetRNA3 predicting transposase-encoding targets. The collected targets for this set of sRNAs (M10, M47, and M62) include over 18 different transposase genes, with all but two interactions showing high probability ≥0.92 (Table S2). This large number of targets results from the

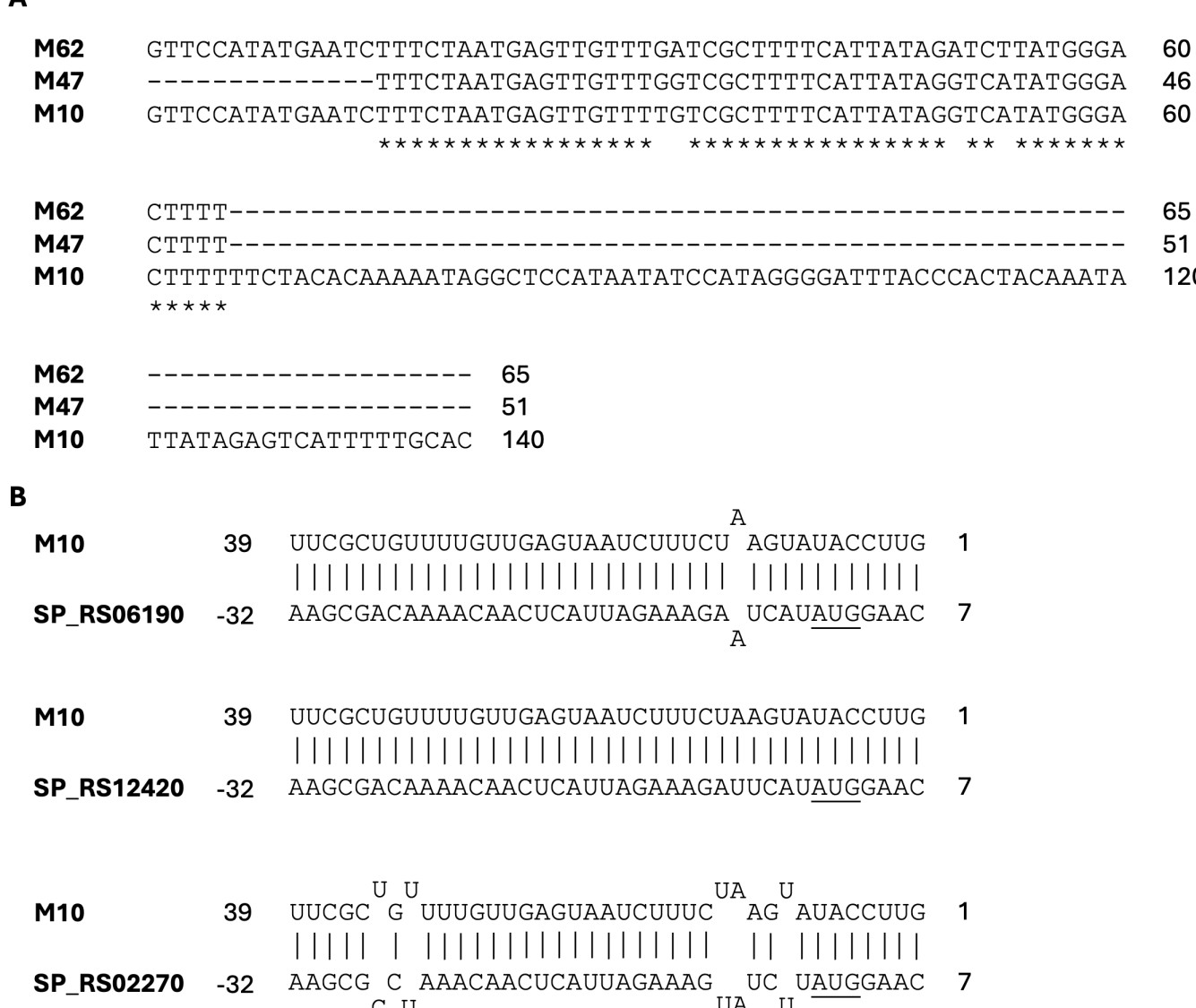

**FIG 4** (A) Clustal omega alignments of M10, M47, and M62, which are predicted to target numerous transposase-encoding transcripts. An asterisk (*) indicates a position in which all three sequences are identical. (B) Three of the M10 interactions predicted by TargetRNA3 in TIGR4. The numbers on either end are relative positions of the interacting sequences within the full RNA sequence. A negative number indicates the sequence is upstream of the mRNA start codon. SP_RS### is the TIGR4 target locus. The underlined "AUGs" are the transposase gene start codons.

duplicated nature of the sequence immediately surrounding the transposase gene (Fig. 4B). However, we note that all but one of the targets of M62 have a predicted probability below our threshold of 0.7, potentially the result of the differences in length between the M62 sequence and the M10 and M47 sequences (region accessibility due to RNA folding is a factor in target prediction and may be influenced by the extra flanking sequence).

There are several well-characterized examples of transposon antisense-encoded RNAs in bacteria, including RNA-OUT, inhibiting IS10 (48), art200, inhibiting IS200 (49), and RNA-C, inhibiting IS30 (50). Transposon-associated antisense RNAs that overlap the transposon coding sequence proximal to the start codon typically act in *trans*, blocking the translation of the transposase (48, 49), but RNA-C, which is antisense to the transposase gene but not directly at the start codon, only appears to act in *cis* (50). Thus, based on the position of these sRNAs proximal and overlapping the ribosome-binding site, it is likely that they are *trans*-acting across the many transposon copies present in the genome. The sequence of M69 is distinct from that of M10, M47, and M62; however, its placement upstream and antisense to an annotated transposase gene suggests a similar functionality.

## *Cis*-encoded sRNA candidates are less common

To identify other *cis*-encoded sRNA candidates, we compared the genomic coordinates of the candidates and MPTs. Candidates suspected to be *cis*-encoded must have genomic coordinates overlapping with the target coordinates. IntaRNA shows 19 candidates may be *cis*-encoded, three of which exhibit probable interaction (Fig. 5A). By contrast, TargetRNA3 suggests only a single candidate to be *cis*-encoded, and it is probable and in common with IntaRNA's results (Fig. 5B). The possible *cis*-encoded sRNA predicted by TargetRNA3 is M66 (Fig. 5C). The M66 sequence appears antisense to the target with perfect binding across 40 nucleotides. M66 targets *ruvB,* which codes for the Holliday junction branch migration DNA helicase RuvB, a subunit in the RuvABC complex. The complex processes Holliday junctions, nucleic acid structures that contain four joined double-stranded arms, during genetic recombination and DNA repair. The individual RuvB subunit is a hexameric ring helicase that acts like a motor to draw the DNA through the complex (51). We believe that the overlap in *cis*-encoded predictions made by TargetRNA3 and IntaRNA suggests that this target is one of interest.

## Seven notable sRNAs for future experimental validation

Across the 58 sRNAs, 7 stood out for reasons that we believe warrant future work to experimentally validate this study's results. Each sRNA is highly conserved and targets a gene with high probability. Four of the notable sRNA candidates share a consensus target between at least two RIP programs (M18, M47, M62, and M66) (Table 4). We believe a consensus target, a gene predicted to be the MPT for an sRNA in at least four of the six *S. pneumoniae* strains, is indicative of a highly likely true sRNA-target pair. Four of the notable candidates target multiple transposases encoding transcripts (M10, M47, M62, and M69). In addition to transposon-associated sRNAs, there are also sRNA candidates with potential metabolic targets such as M63, which has 13 probable targets, including four in the CcpA regulon, and a smattering across the PyrR and CodY regulons. We also checked PneumoBrowse2 (52), an interactive online platform with detailed annotations of *S. pneumoniae* genomes, such as D39V, D39W, and TIGR4, for additional annotations on our seven notable sRNAs. M18 is annotated as part of an anti-toxin/toxin system (Type I addiction module toxin, Fst family), and our predicted target is the mRNA coding for the ClC family H(+)/Cl(-) exchange transporter. From our analysis, we speculate that these seven sRNAs are the most likely to lead to future validation of true sRNA-target pairs that may inform us about the role of sRNAs in *S. pneumoniae* metabolism and virulence.

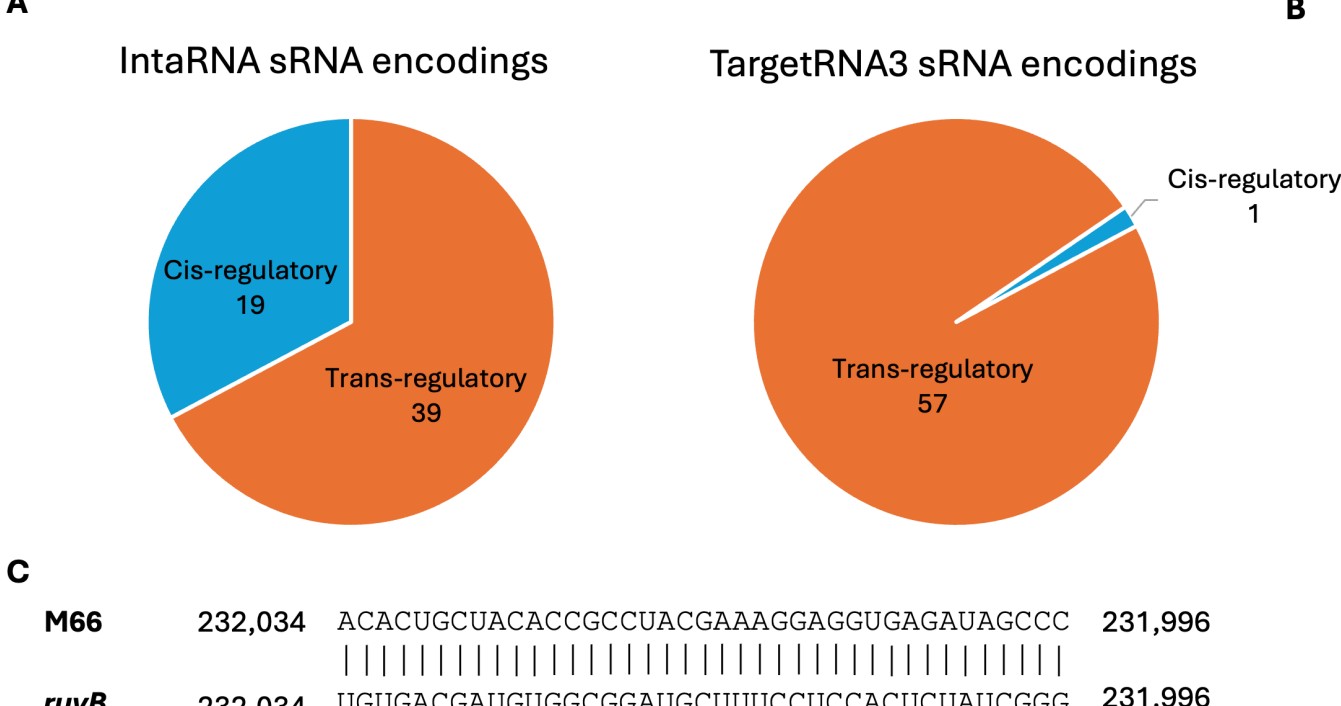

**FIG 5** (A) The distribution of sRNA locations relative to their targets predicted by IntaRNA for 58 candidate sRNAs. (B) The distribution of sRNA locations relative to their targets predicted by TargetRNA3. (C) The M66 cis-interaction is predicted by TargetRNA3, and the sRNA is found on the complement strand and the target on the top strand. The numbers on either end are the exact positions of the interacting sequences in the D39 genome (NC_008533.2).

## Conclusion

This study compiled a list of sRNAs in *S. pneumoniae*, then analyzed their conservation across the Streptococcus genus and within the *S. pneumoniae* pangenome and predicted mRNA targets of conserved sRNAs. BLAST indicates 58 sRNAs exhibit strong conservation across 385 strains. Four RNA-RNA interaction prediction programs made thousands of predictions for the 58 sRNAs. Ultimately, only the probable targets predicted by IntaRNA and TargetRNA3 were the focus of this study's target examination. From our analysis, we propose that there are a handful of transposon-associated sRNAs that target transposases encoding transcripts, likely acting in *trans* based on the position of the base pairing. However, we also identified high-probability targets for other sRNA candidates. For example, M63 (Spd_sr37/srf-04) has several predicted targets across the CcpA regulon, implying a potential role in carbon metabolism. Through this work, we have identified a list of 7 sRNAs for which biological function can be hypothesized. Future work will strive to experimentally validate these hypotheses to reveal more regarding the nature of these sRNAs, their targets in *S. pneumoniae*, and their roles in growth and virulence.

**TABLE 4** Notable sRNAs with a high degree of conservation and interesting predicted targets[a]

| In-house sRNA ID | Other IDs | Consensus | Predicted target(s) | Mechanism | Additional comments |
|---|---|---|---|---|---|
| M63 | Spd_sr37/srf-04 | | CcpA regulon | *trans* | Possibly influences virulence via CcpA regulon |
| M10 | srn266/Spd_sr55 | | Transposases | *trans* | Similar targets to M47, many transposon related |
| M47 | F31 | ✓ | Transposases | *trans* | Similar targets to M10, many transposon related |
| M62 | Spd_sr6/srf-03 | ✓ | Transposases | *trans* | RNA sequence related to M10 and M47 |
| M69 | Spd_sr47 | | Transposases | *trans* | Targets two transposases with high probability |
| M66 | Spd_sr24 | ✓ | *ruvB* | *cis* | *ruvB* involved in recombination and DNA repair |
| M18 | srn503/F67/Spd_sr111 | ✓ | SP_RS05710 | *trans* | Targets an H(+)/Cl(-) exchange transporter |

[a]Other IDs are the labels assigned in the data source studies. The "Consensus" column refers to whether a consensus target was found across multiple RIP programs for the sRNA. The "Mechanism" column refers to whether the sRNA is suspected to be *cis*-acting or *trans*-acting.

## MATERIALS AND METHODS

### Compiling sRNA data sources

Previous studies identified putative sRNAs by high-throughput sequencing in *S. pneumoniae* strains TIGR4 (NC_003028.3) and D39W (CP000410.1). The putative sRNAs in TIGR4 (18, 21) and D39W (22) were narrowed down to a new list of candidates for further analysis of conservation and target prediction (see Results and Discussion).

### Creating the in-house list of putative sRNAs

A list of in-house putative sRNAs was created from the candidates. Previous TIGR4 studies identified different coordinates for transcription initiation and termination sites, so new in-house coordinates were created by combining the smallest initiation and largest termination site coordinates (Fig. S1A). In-house Python scripts retrieved the sRNA sequences from the TIGR4 and D39W genomes using the new coordinates. sRNAs identified in multiple studies under different names were assigned an in-house ID and sequence using the new coordinates (e.g., csRNA5/SN35/srn061/F7/CcnE becomes M1). Differences between the TIGR4 and D39W genomes forced the need to compare sequences rather than coordinates. VectorBuilder ([https://en.vectorbuilder.com/tool/sequence-alignment.html](https://en.vectorbuilder.com/tool/sequence-alignment.html)) aligned sRNAs to confirm the sequences overlap. The largest possible sequence became the new in-house sRNA by joining the overlapping subsequence and trailing sequences on either end (Fig. S1B). After forming the list of in-house sRNAs, sequences with a length of <50 nucleotides were removed.

### Assessing conservation of sRNAs

The in-house sRNAs were aligned to the genomes of 385 *S. pneumoniae* strains (25, 26), *S. pyogenes* (NZ_LS483338.1), *S. mutans* (NZ_CP044221.1), *S. suis* (NC_012926.1), *S. mitis* (NZ_GL397179.1), *S. oralis* (NZ_LR134336.1), and *S. gordonii* (NZ_CP077224.1). Raw reads and the mapping results for 350 *S. pneumoniae* strains (25) available in BAM format were converted to consensus files in FASTA format with the samtools consensus mode (53). Then, a database, created with makeblastdb, containing the sRNA sequences was aligned to each genome using blastn with the task parameter set to megablast. Similarly, a database created with makeblastdb containing the 385 *S. pneumoniae* strains was aligned to each of the sORF nucleotide and peptide sequences using blastn and tblastn. The e-value maximum threshold was raised from a default of 10–20 for the tblastn alignments. An in-house Python script retrieved the average number of alignments, best sequence identity, and number of genomes with an alignment for each sRNA. In-house sRNAs with an average number of alignments per strain >15 were classified as potentially highly repetitive sequences and removed from the in-house list. sRNAs not appearing in the majority of the 385 strains were also removed. A database containing the *S. pyogenes*, *S. mutans*, *S. suis*, *S. mitis*, *S. oralis*, and *S. gordonii* genomes, created with makeblastdb, was used to search for sRNA sequence alignments using blastn with the task parameter set to blastn. Synteny of the sRNAs was evaluated by comparing the 1,000 nt upstream and downstream of the sequences to those in TIGR4. The 1,000 nt on either end of the sRNA sequences for the other 384 strains were compared base-wise to obtain sequence identities. The average sequence identities, both up and downstream, were averaged across all 384 strains, and if either average sequence identity, up or downstream, is ≥75% then it was concluded that the synteny of the sRNA is preserved.

### sRNA target prediction

IntaRNA version 3.3.2, sRNARFTarget, and TargetRNA3 were used to predict targets for the 58 sRNA candidates. CopraRNA version 2.1.4 was only used to predict targets for three sRNAs with significant alignments in *S. suis*, *S. pyogenes*, *S. mutans*, *S. mitis*, *S. oralis*, and *S. gordonii*. Predictions for each sRNA were made in six different *S. pneumoniae* strains consisting of TIGR4 (NC_003028.3), D39 (NC_008533.2),

TVO_Taiwan19F-14, TVO_1901920, TVO_1901934, and TVO_1902277 (26). IntaRNA was run with the IntaRNAsTar personality, the number of predictions set to 5, and otherwise default parameters. sRNARFTarget was run with the provided Docker container. sRNARFTarget requires a user input transcriptome, so an in-house Python script created two transcriptome files for each of the six *S. pneumoniae* strains using the coordinates from the respective GenBank files and retrieving the sequences from the genome. The two transcriptomes are defined by the exact gene coordinates and coordinates adjusted to include 100 nucleotides upstream of the start codon and 300 nucleotides downstream of the stop codon or until the next gene's start codon, whichever comes first. TargetRNA3 was run with the probability threshold lowered to 0.25 and otherwise default parameters. Note, the six genomes were first added to the local user database by providing the accession identifier. CopraRNA was run with default parameters in *S. pneumoniae* (NC_003028), *S. suis* (NC_012926), *S. pyogenes* (NZ_LS483338), *S. mutans* (NZ_CP044221), *S. mitis* (NZ_CP012646), *S. oralis* (NZ_LR134336), and *S. gordonii* (NZ_CP077224).

## sRNA target analysis

An in-house Python script retrieved the sequences including 25 nucleotides upstream and downstream of the mRNA interacting sequence. The structures of these sequences were predicted using RNAfold version 2.6.4 with default parameters. If the sRNA interacting sequence overlaps with a structured region of the mRNA, then the interaction was labeled as interacting with a structured region. Only the MPTs of each sRNA in the six strains predicted by IntaRNA were analyzed. To determine whether the sRNAs are acting in regulons, we searched the target loci against the RegPrecise database (https://regprecise.lbl.gov/index.jsp). Only the MPTs and probable targets predicted by TargetRNA3 in TIGR4 were searched, seeing as TIGR4 is the only *S. pneumoniae* strain for which the database contains information. The expression of sRNA sequences was confirmed by checking if the transcription initiation and termination sites were present in 5′ and 3′ RNA-end sequencing data (39).

## ACKNOWLEDGMENTS

M.C.E. was partially funded by a Boston College Undergraduate Research Fellowship and by grants R21AI181123 and R01GM134259 from the US National Institutes of Health.

We thank Quinlan Furumo for careful copyediting of the manuscript.

M.C.E. roles: data curation, investigation, visualization, writing—original draft. M.M.M. roles: conceptualization, data curation, funding acquisition, supervision, writing—review and editing.

## AUTHOR AFFILIATION

[1]Department of Biology, Boston College, Chestnut Hill, Massachusetts, USA

## AUTHOR ORCIDs

Michelle M. Meyer  http://orcid.org/0000-0001-7014-9271

## FUNDING

| Funder | Grant(s) | Author(s) |
| --- | --- | --- |
| National Institutes of Health | R21AI181123,R01GM134259 | Michelle M. Meyer |

## AUTHOR CONTRIBUTIONS

Matthew C. Eichelman, Data curation, Investigation, Visualization, Writing – original draft | Michelle M. Meyer, Conceptualization, Data curation, Funding acquisition, Supervision, Writing – review and editing

## ADDITIONAL FILES

The following material is available online.

## Supplemental Material

**Data File S1 (Spectrum03252-24-s0001.xlsx).** List of source putative sRNAs.

**Data File S2 (Spectrum03252-24-s0002.xlsx).** List of curated sRNAs.

**Data File S3 (Spectrum03252-24-s0003.xlsx).** CopraRNA predicted targets.

**Data File S4 (Spectrum03252-24-s0004.xlsx).** IntaRNA predicted targets.

**Data File S5 (Spectrum03252-24-s0005.xlsx).** sRNAFTarget predicted targets (coding regions).

**Data File S6 (Spectrum03252-24-s0006.xlsx).** sRNAFTarget predicted targets (coding and flanks).

**Data File S7 (Spectrum03252-24-s0007.xlsx).** TargetRNA predicted targets.

**Supplemental figures and tables (Spectrum03252-24-s0008.pdf).** Fig. S1, and Tables S1 and S2.

## Open Peer Review

**PEER REVIEW HISTORY (review-history.pdf).** An accounting of the reviewer comments and feedback.

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
