## [Reviewer comments · Microbiology Spectrum]

Microbiology Spectrum

Assessing the conservation and targets of putative sRNAs in *Streptococcus pneumoniae*

Matthew Eichelman and Michelle Meyer

Corresponding Author(s): Michelle Meyer, Boston College

Review Timeline:

Submission Date:	December 14, 2024
Editorial Decision:	January 20, 2025
Revision Received:	March 19, 2025
Editorial Decision:	April 22, 2025
Revision Received:	April 25, 2025
Accepted:	April 27, 2025

Editor: Tino Polen

Reviewer(s): The reviewers have opted to remain anonymous.

Transaction Report:

DOI: <https://doi.org/10.1128/spectrum.03252-24>

Re: Spectrum03252-24 (Assessing the conservation and targets of putative sRNAs in *Streptococcus pneumoniae*)

Dear Professor Michelle M Meyer,

thank you for submitting your manuscript to Microbiology Spectrum.

I received comments from two expert reviewers on your manuscript.

While both reviewers recognize your manuscript as a valuable computational study of putative sRNA targets in *S. pneumoniae*, they also suggest serious modifications to improve the manuscript. I do hope you find the reviewers' comments below helpful and are able to provide a revised version of your manuscript.

Revision Guidelines

Sincerely,
Tino Polen
Editor
Microbiology Spectrum

Reviewer #1 (Comments for the Author):

This is a valuable computational study that applies analysis criteria to begin to understand possible functions of sRNAs *Streptococcus pneumoniae* identified previously in several studies. One criterion is conservation across a wide range of different

serotype and subgroups of *S. pneumoniae* as well as evidence of expression by northern blotting or RT-qPCR. Certain annotated sequences, such as BOX elements and other RNA motifs, were also eliminated. This led to an initial list of 76 conserved, non-repetitive sRNAs. Synteny conservation in chromosomes narrowed the list down to 58 sRNA candidates for further analysis by several RNA-RNA interaction prediction (RIP) programs. Three general conclusions came out of this analysis. The RIP programs successfully identified known targets for the csRNAs, which had been studied previously, thereby validating the general analysis. However, the RIP programs predicted thousands of putative sRNA-target pairs and therefore, were not informative in many cases. Nonetheless, the RIP analysis identified seven highly conserved sRNAs that showed strong putative interaction with certain mRNA targets. The study discusses these targets and possible functions very briefly and suggests that these seven putative interactions will be a good starting place to better understand the biology of sRNA regulation in *S. pneumoniae*.

As stated above, these computational results are valuable in that they identify some strongly conserved candidate sRNA-mRNA interactions and that they reveal the limitations of the RIP programs for identifying targets for most sRNAs in *S. pneumoniae*. However, there are a couple major issues with this manuscript. First, the figures could be considerably more informative and integrated. Second, along with the figures, more context about the biology of the seven putative sRNA targets could be included. Most importantly, the manuscript is somewhat poorly organized and edited. It seems like this version of the manuscript was a shortened version of a previous longer version. Many of the Table and Figure citations in the text do not correspond to correct Tables and Figures. These and some other comments follow.

1. Line 48. Change to: "Gram-positive". Please check throughout.
2. Line 55. This refers only to the USA and not worldwide, where the impact is much greater. Please clarify and expand.
3. Line 66. Please cite the review by Olejniczak, Jiang, Basczok, and Storz on possible other RNA matchmakers in Gram-positive bacteria (doi: 10.1111/mmi.14842) or other recent reviews about possible RNA chaperones in Gram-positive bacteria.
4. Line 69. Delete: "extremely."
5. Line 86. The sRNAs do not interact with the "transposases" (which are proteins) as stated. Please correct and check for other instances of shorthand that leads to misattribution.
6. Figure 1 and throughout. Some of the type sizes in this and other figures are too small to read easily. Also, the white type against pale backgrounds is hard to read. Please fix.
7. Line 127. The other *Streptococcus* species mentioned (*pyogenes*, *mutans*, and *suis*) are evolutionarily distant from *S. pneumoniae*. It would be highly informative to extend the analysis to other closer relatives in the *Streptococcus* species in the mitis group, such as *S. mitis*, *oralis*, *gordonii*, etc.
8. Table 1. The table would be more informative if the 58 final candidates were highlighted.
9. Line 204. Is a two-nucleotide interaction sufficient and significant?
10. Line 205. Please provide an argument for this "belief."
11. Line 211. The citations to Tables and Figures start to fall apart here. Table 3 focuses on M63, which is not the subject of this paragraph. You seem to mean Table 2 that summarizes the results discussed in this paragraph.
12. Line 217. Describe the known targets associated with virulence. This could be added to Table 2. In addition, which transporters are regulated by M23? Descriptions of biological roles would be much clearer if they were included in Table 2 and in other later Tables.
13. Line 232. These results are listed in Table 3. Please cite.
14. Line 239. Don't you mean Figure 3a?
15. Line 240. Don't you mean Table 3? There is no Table 4.
16. Line 267. Please change to: "Notably".
17. Lines 270-274 and Figure 3. This section would benefit from a summary diagram beyond what is presented in Figure 3.
18. Line 277. Start new paragraph after "prediction."
19. Line 293. A header is needed for this new section.

20. Line 325. Include: "D39W", since it is mentioned with regard to one of the studies and is the PneumoBrowse2.

21. Can the targets in Table 2 and 3 be integrated with the 7 putative candidates in Table 4. For example, what are the possible biological implications of the different CcpA regulon targets and their regulation by different sRNAs? What are the different transposase functions and how are these targets related?

Reviewer #2 (Comments for the Author):

In this manuscript entitled "Assessing the conservation and targets of putative sRNAs in *Streptococcus pneumoniae*", Eichelman and Meyer use a set of rationale and bioinformatic tools to predict which of the previously identified small RNAs may be small regulatory RNAs (sRNAs) (Fig. 1), i.e., small RNAs that regulate gene expression by base-pairing with mRNAs. Furthermore, the authors determine the conservation of these putative sRNAs among strains of *S. pneumoniae* (Fig. 2, Table 1). The authors then use a set of previously described sRNA target prediction programs to identify which mRNAs, particular sRNA candidates are likely to base-pair with and consequently regulate. The authors then designate mRNAs as the most probable target of particular sRNAs based on multiple programs predicting a high probability of pairing between the sRNA and that target. From these analyses the authors come up with a list of likely mRNA targets of a set of sRNAs (Tables 2, 3, and 4 and Figs 3, 4 and 5), which the authors feel are a high priority to investigate in the future.

In short, this manuscript is interesting and has the potential to become an important paper. The importance of this manuscript lies in the fact that little is known about sRNA-mediated gene regulation in *S. pneumoniae*. Out of the hundreds of small RNAs identified so far in *S. pneumoniae* only five, the Ccn sRNAs, have been established as regulating gene expression through base-pairing with mRNAs and even for those sRNAs, we likely only know a small subset of their true targets. But, in my opinion, this manuscript needs improvement. Importantly, in this current manuscript version, the authors do not take into account some of the crucial, existing knowledge regarding some of these pneumococcal small RNAs and some important properties of sRNAs established in other organisms that are likely properties of sRNAs in the pneumococcus. First of all, prior work by Michael Federle's group using ribosome profiling has established that many of the small RNAs of interest including M63 are translated by the ribosome in *S. pneumoniae* and are likely small mRNAs (see comments #2, #3, and #4). While so-called dual function regulators, i.e., RNAs that serve both regulatory and mRNA functions are known to exist in other bacteria, the authors should incorporate established data of the protein encoding capacity of the candidate small RNAs and the conservation of their ORFs in their analyses. Additionally, the authors seem to ignore certain principles derived from studies of sRNAs in other bacterial species, such as the finding that sequences of the sRNA involved in base-pairing are usually single stranded (comment #5). Structure in the mRNA target is a little less important as the sRNA may pair with the mRNA before the sequence involved in intramolecular base-pairing in the target mRNA emerges from RNA polymerase during transcription. On a minor note, the authors should double check that the figure/table that they reference is the correct one, as there seems to be multiple instances where the incorrect figure is referenced.

Major criticisms

1. L94-97. The authors state that they winnowed their list of sRNAs by including only sequences identified in multiple studies, yet according to Fig. 1D, many of the sRNAs on this list appear in only one of the three studies. The authors need to address this apparent inconsistency in their statement and the figure.
2. L90-103. The authors develop a list of sRNAs based on several different criteria. Notably missing from this analysis is whether or not the putative sRNA encodes a protein. Many of the sRNAs that made it to the authors final sRNA list (additional data file 2) have been previously shown to be translated (see PMID 35852327). So, it is possible that many of these are not sRNAs, but small mRNAs, although it is also possible that they have both functions, e.g., like SgrS, Spot42, or AzuCR in *E. coli*. Regardless, I think that the authors need to address this issue.
3. Along those same lines, the authors could include in Table 1, whether or not the "sRNA" has been shown to encode a small protein, whether there is a potential small ORF contained within the sRNA, and whether the amino acid sequence of this putative ORF is conserved.
4. The authors emphasize the potential regulation of mRNAs by M63, but it remains possible that M63 is a small mRNA. M63 appears to encode rio16, an 11 amino acid protein. Is this orf conserved among the genomes of the *S. pneumoniae* strains analyzed.
5. L232-249. I would say that a fundamental principle that has been established for sRNAs in *E. coli* or *S. enterica* is that the regions of the sRNA involved in initiating base-pairing, the so called "seed sequences" are single stranded. If the seed sequence is paired with another portion of the sRNA, it is not available to base-pair with target mRNAs. Thus, it is plausible that M63 could potentially base-pair with hexA. But, in my opinion, if the RNAfold prediction reflects reality, then I am skeptical that M63 would interact with adhE or glgB (Fig. 3). While I could be wrong, this idea of pairing sequences being in single stranded regions for

known sRNAs, and how this knowledge impacts one's consideration of the potential for target interactions to be "real" should be highlighted and discussed.

Minor comments

L15. Remove the apostrophe in "sRNA's" that indicates possession.

L17. Change ", however" to "; however,"

L40. I do not understand what the authors are trying to convey when they state "to what extent these candidates may overlap". Overlap how?

L41. Change "focus" to plural.

L44. Sentence needs rewriting. Perhaps, change "reveal more about the" to "have a"

L50-52. This is an underestimate of infection and deaths due to pneumococcal disease. The number of deaths is closer to 1-2 million deaths per year. (see PMID 30243584).

L56-57. It may be helpful for the authors to provide context to this statement. The reason people tend to focus on sRNAs encoded in intergenic regions is because they are easier to identify and because an RNA transcribed in an intergenic region that lacks an ORF is deemed more likely to be a functional regulatory RNA than a portion of a protein, tRNA, or rRNA encoded transcript that could potentially be an useless, intermediate in RNA decay.

L85-87. The sRNAs do not interact with transposases or the gene involved in carbon metabolism, but rather their respective mRNAs.

L169. Change to "sequences that we examined"

Line 194, Change to past tense, "observed"

L211. Table 3 is reference, but it is not clear why. Perhaps the authors mean Table 2?

L274. Figure 3 should be Figure 4.

L286. Change to "; however,"

Eichelman and Meyer Response to Reviewers

Reviewer #1 (Comments for the Author):

This is a valuable computational study that applies analysis criteria to begin to understand possible functions of sRNAs *Streptococcus pneumoniae* identified previously in several studies. One criterion is conservation across a wide range of different serotype and subgroups of *S. pneumoniae* as well as evidence of expression by northern blotting or RT-qPCR. Certain annotated sequences, such as BOX elements and other RNA motifs, were also eliminated. This led to an initial list of 76 conserved, non-repetitive sRNAs. Synteny conservation in chromosomes narrowed the list down to 58 sRNA candidates for further analysis by several RNA-RNA interaction prediction (RIP) programs. Three general conclusions came out of this analysis. The RIP programs successfully identified known targets for the csRNAs, which had been studied previously, thereby validating the general analysis. However, the RIP programs predicted thousands of putative sRNA-target pairs and therefore, were not informative in many cases. Nonetheless, the RIP analysis identified seven highly conserved sRNAs that showed strong putative interaction with certain mRNA targets. The study discusses these targets and possible functions very briefly and suggests that these seven putative interactions will be a good starting place to better understand the biology of sRNA regulation in *S. pneumoniae*.

We thank the reviewers for their positive view of this work.

As stated above, these computational results are valuable in that they identify some strongly conserved candidate sRNA-mRNA interactions and that they reveal the limitations of the RIP programs for identifying targets for most sRNAs in *S. pneumoniae*. However, there are a couple major issues with this manuscript.

First, the figures could be considerably more informative and integrated.

We have substantially revised the figures to be more informative and reflective of the results. Specifically Figures 3 and 4 have changed substantially to better illustrate the biologically relevant targets of the sRNAs.

Second, along with the figures, more context about the biology of the seven putative sRNA targets could be included.

We have extended our discussion of seven putative sRNA candidates to give more biological context for their importance. In particular we have replaced Figure 3 with one that better demonstrates the interactions between M63 and various predicted target regions. We also pulled former supplemental figures into the main text (Figure 4) showing the alignment of three sRNAs predicted to target transposon transcripts (M10, M47, and M62) and the predicted pairings of M10 to mRNA regions that include the transposase start codons.

Most importantly, the manuscript is somewhat poorly organized and edited. It seems like this version of the manuscript was a shortened version of a previous longer version. Many of the

Table and Figure citations in the text do not correspond to correct Tables and Figures. These and some other comments follow.

We have carefully gone through the manuscript to remove as many of these issues as we could identify.

1. Line 48. Change to: "Gram-positive". Please check throughout.

Changed "Gram positive" to "Gram-positive". Also checked throughout and did not find any other instances.

2. Line 55. This refers only to the USA and not worldwide, where the impact is much greater. Please clarify and expand.

Added "in the United States" and a sentence immediately after describing the impact worldwide. The new sentence reads

"Streptococcus pneumoniae is a Gram-positive bacterium that causes various diseases including pneumonia, meningitis, bacteremia, otitis media and sinusitis. Invasive pneumococcal disease is particularly dangerous in children and the elderly (CDC et al., 2013), and in 2004 was responsible for approximately 4 million illness episodes, 445,000 hospitalizations, and 22,000 deaths in the United States (Huang et al., 2011). In 2016, S. pneumoniae was the leading cause of lower respiratory infection morbidity and mortality globally causing over a million deaths (GBD 2016 Lower Respiratory Infections Collaborators, 2016)."

3. Line 66. Please cite the review by Olejiczak, Jiang, Basczok, and Storz on possible other RNA matchmakers in Gram-positive bacteria (doi: 10.1111/mmi.14842) or other recent reviews about possible RNA chaperones in Gram-positive bacteria.

We have added additional references to discuss possible RNA matchmakers in Gram-positive bacteria. This section now reads:

"However, the role of RNA chaperones in Gram-positive bacteria is substantially less clear. Such proteins are frequently not present, and when they are present their functionality is often substantially different from that observed in gram-negative organisms (Christopoulou and Granneman, 2022). In S. pneumoniae, recent work shows KH domain proteins, like KhpA and KhpB, are associated with sRNAs (Olejiczak et al., 2022), and sRNAs have been associated with the exonuclease CbfI (Hor et al. 2020). However, there are no confirmed sRNA chaperone homologs in the S. pneumoniae genome (Zhang et al., 2003)."

4. Line 69. Delete: "extremely."

Removed "extremely". The sentence now reads "but follow-up characterization has been limited."

5. Line 86. The sRNAs do not interact with the "transposases" (which are proteins) as stated. Please correct and check for other instances of shorthand that leads to misattribution.

Changed "interact with" to "interact with mRNAs encoding". The sentence now reads "The predictions suggest 7 putative sRNAs, which have highly probable targets, interact with mRNAs coding for transposases and genes involved in carbon metabolism regulation." We searched the manuscript for other such instances and did not find any other instances of shorthand leading to misattribution, however, language was changed throughout to clarify such instances.

6. Figure 1 and throughout. Some of the type sizes in this and other figures are too small to read easily. Also, the white type against pale backgrounds is hard to read. Please fix.

Changed all white text in the figures to black to make it easier to read. Changed all figures to have a font size of 8 or greater throughout.

7. Line 127. The other Streptococcus species mentioned (pyogenes, mutans, and suis) are evolutionarily distant from *S. pneumoniae*. It would be highly informative to extend the analysis to other closer relatives in the Streptococcus species in the mitis group, such as *S. mitis*, *S. oralis*, *S. gordonii*, etc.

*We extended the conservation analysis to *S. mitis*, *S. oralis*, and *S. gordonii* and revised the section to focus on the sRNAs highly conserved across all 6 species (*S. pyogenes*, *S. mutans*, *S. suis*, *S. mitis*, *S. oralis*, *S. gordonii*). The text now reads:*

*"To determine if any of the sRNAs are conserved in species other than *S. pneumoniae* we also analyzed a range of other species in the Streptococcus genus. Only 2 of the candidate sRNAs (M77 and M81) align with sequence identities >65% to each of *S. pyogenes*, *S. mutans*, *S. suis*, *S. mitis*, *S. oralis*, and *S. gordonii*, and 25 additional sRNA candidates can be identified in a subset of these organisms (Supplemental Table S1). Thus, it appears over half of the sRNAs are unique to *S. pneumoniae*"*

8. Table 1. The table would be more informative if the 58 final candidates were highlighted.

The final 58 candidates are now highlighted in the table with the other 18 candidates greyed.

9. Line 204. Is a two-nucleotide interaction sufficient and significant?

We agree that 2 nucleotides of pairing are unlikely to represent a stable pairing, such pairings in isolation are typically not returned in isolation by RNAfold due to their low stability, and more often occur within the context of additional predicted structure. We considered 2 consecutive pairs as the minimum threshold within the local region that may be disrupted upon interaction with the sRNA. We have substantially changed this section to increase clarity based on other reviewer comments, the revised paragraph now reads:

"In many cases sRNA pairing can impact the internal base pairing of the mRNA to enable gene expression changes (Papenfert and Storz 2024). Thus, we assessed whether the putative mRNA

target regions are likely to have internal base pairing that may be impacted sRNA interaction. We used RNAfold (Lorenz et al., 2011) to predict the structure of mRNA target sequences including 25 nucleotides up/downstream of the binding region in the absence of any sRNA partner. We consider a structured region to be a segment of the mRNA sequence displaying internal base pairing (>2 consecutive bases). Across the MPTs predicted by IntaRNA and TargetRNA3, 51 out of 58 sRNAs base pair with a region considered structured suggesting that the putative sRNA-mRNA interaction may induce structural changes in the secondary or tertiary structures to enable regulation (Tieng et al., 2023).'

10. Line 205. Please provide an argument for this "belief."

As noted above, we have substantially changed this section of the manuscript (see comment above) to remove this language and clarify the evidence supporting the potential for sRNAs-interaction to modify mRNA internal structure.

11. Line 211. The citations to Tables and Figures start to fall apart here. Table 3 focuses on M63, which is not the subject of this paragraph. You seem to mean Table 2 that summarizes the results discussed in this paragraph.

Changed "Table 3" to "Table 2". All subsequent typographical errors leading to confusion have been fixed.

12. Line 217. Describe the known targets associated with virulence. This could be added to Table 2. In addition, which transporters are regulated by M23? Descriptions of biological roles would be much clearer if they were included in Table 2 and in other later Tables.

We have expanded and clarified our discussion of the sRNA targets associated with virulence. The experimentally determined targets of the csRNAs (M1) associated with virulence were discussed in more detail above during our assessment of the prediction algorithms. Unfortunately, the putative target associated with F25, F21, F44 in Mann et al. (SP_RS08340) is not well-characterized, but we have added what description is available to our text and the legend for Table 2.

"Previous studies used transposon insertion mutants to conclude specific sRNAs may support virulence in TIGR4 (Mann et al., 2012). We compared our predicted targets for these sRNAs to evaluate these hypotheses. Previous work suggested 8 putative sRNAs (Table 2) play a definitive role in pathogenesis, and some individual target loci were identified by microarray analysis of attenuated sRNA mutants (Mann et al., 2012). Only 3 of these sRNAs met our criteria for further investigation (Fig. 2A, Table 2). The others overlap with known cis-regulatory elements (F20 and F44), transfer-messenger RNA (F32), or were removed following conservation analysis (F41, F48). F41 is one of the sRNAs found in <12 strains, and F48 was deemed a repetitive sequence (average of 36 copies per genome). Among the three remaining candidates, one (M1/F7) is a csRNA with an established role in pathogenesis (DeLay 2024, Patenge et al., 2012, Schnorpfeil et al., 2013). The other two, M45 and M23, are not characterized. M45 is predicted to target type IV teichoic acid flippase TacF that is responsible for transporting choline across the cytoplasmic membrane, a nutritional requirement of S.

pneumoniae (Damjanovic et al., 2007). M23 was originally reported to target SP_RS08340-50, a putative carbohydrate transporter, based on microarray analysis, but we predict that it targets a transposase encoding transcript (SP_RS13320) (Table 2). Without validating the targets, the role of M45 and M23 in pathogenesis is unclear; however the predicted target (*tacF*) of M45 is suggestive of such a role.”

“**Table 2:** A comparison of the in-house target predictions and the putative targets identified by microarray analysis. The previously identified putative targets SP_RS08340-50 are three neighboring loci involved in carbohydrate transport (carbohydrate ABC transporter permease) and proposed to be collectively regulated by three of the sRNAs (Mann et al., 2012).”

13. Line 232. These results are listed in Table 3. Please cite.

Added “(Table 3)” at the end of the sentence to cite the table.

14. Line 239. Don't you mean Figure 3a?

Yes, we changed “Figure 4a” to “Fig. 3A” and corrected all similar mistakes.

15. Line 240. Don't you mean Table 3? There is no Table 4.

Changed to “Table 4” to “Table 3”.

16. Line 267. Please change to: "Notably".

Changed “Notable” to “Notably”.

17. Lines 270-274 and Figure 3. This section would benefit from a summary diagram beyond what is presented in Figure 3.

We moved Supplementary Figure S2 (depicting the alignments of M10, M47, and M62) to the main text and labeled as Fig. 4A. The original Figure 4 is now Figure 4B. We also added a supplemental table (Supplemental Table 2) to fully elaborate all the predicted targets transposase associated targets for these sRNA candidates.

18. Line 277. Start new paragraph after "prediction."

Started a new paragraph after “(target prediction).”.

19. Line 293. A header is needed for this new section.

Added a header for the new section that reads “Cis-encoded sRNA candidates are less common”.

20. Line 325. Include: "D39W", since it is mentioned with regard to one of the studies and is the PneumoBrowse2.

Added D39W to the list of genomes available on PneumoBrowse.

21. Can the targets in Table 2 and 3 be integrated with the 7 putative candidates in Table 4. For example, what are the possible biological implications of the different CcpA regulon targets and their regulation by different sRNAs? What are the different transposase functions and how are these targets related?

We added a column to Table 4 to summarize the functional targets and their biological significance. While we appreciate that adding the specific targets from Tables 2 and 3 might consolidate it all onto one table, the variety of different information types needed for each sRNA make this challenging, and we feel ultimately detract from the emphasis that we want to put on these particular sRNAs vs. all of those presented on Tables 2 and 3.

Reviewer #2 (Comments for the Author):

In this manuscript entitled "Assessing the conservation and targets of putative sRNAs in *Streptococcus pneumoniae*", Eichelman and Meyer use a set of rationale and bioinformatic tools to predict which of the previously identified small RNAs may be small regulatory RNAs (sRNAs) (Fig. 1), i.e., small RNAs that regulate gene expression by base-pairing with mRNAs. Furthermore, the authors determine the conservation of these putative sRNAs among strains of *S. pneumoniae* (Fig. 2, Table 1). The authors then use a set of previously described sRNA target prediction programs to identify which mRNAs, particular sRNA candidates are likely to base-pair with and consequently regulate. The authors then designate mRNAs as the most probable target of particular sRNAs based on multiple programs predicting a high probability of pairing between the sRNA and that target. From these analyses the authors come up with a list of likely mRNA targets of a set of sRNAs (Tables 2, 3, and 4 and Figs 3, 4 and 5), which the authors feel are a high priority to investigate in the future.

In short, this manuscript is interesting and has the potential to become an important paper. The importance of this manuscript lies in the fact that little is known about sRNA-mediated gene regulation in *S. pneumoniae*. Out of the hundreds of small RNAs identified so far in *S. pneumoniae* only five, the Ccn sRNAs, have been established as regulating gene expression through base-pairing with mRNAs and even for those sRNAs, we likely only know a small subset of their true targets. But, in my opinion, this manuscript needs improvement. Importantly, in this current manuscript version, the authors do not take into account some of the crucial, existing knowledge regarding some of these pneumococcal small RNAs and some important properties of sRNAs established in other organisms that are likely properties of sRNAs in the pneumococcus. First of all, prior work by Michael Federle's group using ribosome profiling has established that many of the small RNAs of interest including M63 are translated by the ribosome in *S. pneumoniae* and are likely small mRNAs (see comments #2, #3, and #4). While so-called dual function regulators, i.e., RNAs that serve both regulatory and mRNA functions are known to exist in other bacteria, the authors should incorporate established data of the protein encoding capacity of the candidates small RNAs and the conservation of their ORFs in their analyses. Additionally, the authors seem to ignore certain principles derived from studies of sRNAs in other bacterial species, such as the finding that sequences of the sRNA involved in base-pairing

are usually single stranded (comment #5). Structure in the mRNA target is a little less important as the sRNA may pair with the mRNA before the sequence involved in intramolecular base-pairing in the target mRNA emerges from RNA polymerase during transcription. On a minor note, the authors should double check that the figure/table that they reference is the correct one, as there seems to be multiple instances where the incorrect figure is referenced.

We thank the reviewer for their generally positive view and thank them for pointing us toward the sORF work of Federle's group, which was not properly incorporated into the work previously. We have also clarified some of discussion of potential mRNA-sRNA pairing regions to more clearly articulate the nature of the predicted pairings throughout.

Major criticisms

1. L94-97. The authors state that they winnowed their list of sRNAs by including only sequences identified in multiple studies, yet according to Fig. 1D, many of the sRNAs on this list appear in only one of the three studies. The authors need to address this apparent inconsistency in their statement and the figure.

We clarified this section, the sRNA had to meet one of the three criteria. The section now reads:

"We further narrowed this pool to a list where each sRNA has at least 1 of 3 attributes: 1) sequence identified in multiple studies, 2) expression confirmed by Northern blot or RT-qPCR, 3) sequence characterized as a csRNA or overlapping with a sORF (Laczkovich et al., 2022)"

2. L90-103. The authors develop a list of sRNAs based on several different criteria. Notably missing from this analysis is whether or not the putative sRNA encodes a protein. Many of the sRNAs that made it to the authors final sRNA list (additional data file 2) have been previously shown to be translated (see PMID 35852327). So, it is possible that many of these are not sRNAs, but small mRNAs, although it is also possible that they have both functions, e.g., like SgrS, Spot42, or AzuCR in E. coli. Regardless, I think that the authors need to address this issue.

We have added a subsection to the results section to explicitly discuss the overlap between sORFs and the candidate sRNAs, this includes an explicit discussion and citation of dual-function sRNA/sORFs in other species.

"Several sRNA candidates overlap with reported sORFs

Of the 58 final sRNAs candidates, 6 of them overlap with previously identified sORFs (Table 1) (Laczkovich et al., 2022), and one (M64) is complementary to an sORF. Moreover, we see that these 5 of the 7 sORFs have 1 alignment per genome and a high degree of conservation with nucleotide and peptide sequence identities >98% and are found in nearly all 385 strains. The first exception, rio56, is a 6 amino acid peptide sequence and is too short for BLAST to produce alignments with an e-value < 20, and the second, rio86, has an average peptide sequence identity of 93.6%. Of particular interest is M61 (Spd_sr5/srf-02), which overlaps with rio3, whose expression was shown to promote in vivo fitness (Laczkovich et al., 2022). However, dual-function RNAs, sequences with coding and non-coding functions, are known to exist in other

bacteria like SgrT/SgrS and AzuC/AzuR in Escherichia coli (Aoyama and Storz, 2023) and Pel RNA in Streptococcus pyogenes (Raina et al., 2018). Thus, we decided to retain sORF overlapping sRNAs for target analysis.”

3. Along those same lines, the authors could include in Table 1, whether or not the "sRNA" has been shown to encode a small protein, whether there is a potential small ORF contained within the sRNA, and whether the amino acid sequence of this putative ORF is conserved.

“We added a column in Table 1 indicating whether an sRNA overlaps with a sORF and as noted above added a section to the results to explicitly discuss this. We find that the sORFs are generally all well-conserved, and this is described in the section added above.”

4. The authors emphasize the potential regulation of mRNAs by M63, but it remains possible that M63 is a small mRNA. M63 appears to encode rio16, an 11 amino acid protein. Is this orf conserved among the genomes of the S. pneumoniae strains analyzed.

We discuss the overlap between M63 and rio16 in the section that calls out the collective potential targets of M63.

“M63 also overlaps with the rio16 sORF, indicating that part of this sRNA is translated. The sORF has a conserved sequence with average sequence identity of 99.7% across the pangenome. However, it remains possible that M63 is a dual-function RNA both encoding a small protein and regulating the members of the CcpA regulon.”

5. L232-249. I would say that a fundamental principle that has been established for sRNAs in E. coli or S. enterica is that the regions of the sRNA involved in initiating base-pairing, the so called "seed sequences" are single stranded. If the seed sequence is paired with another portion of the sRNA, it is not available to base-pair with target mRNAs. Thus, it is plausible that M63 could potentially base-pairing hexA. But, in my opinion, if the RNAfold prediction reflects reality, then I am skeptical that M63 would interact with adhE or glgB (Fig. 3). While I could be wrong, this idea of pairing sequences being in single stranded regions for known sRNAs, and how this knowledge impacts one's consideration of the potential for target interactions to be "real" should be highlighted and discussed.

Based on the feedback of both reviewers, we feel like the previous Figure 3 over-represented our confidence in the RNAfold structure, and its importance in our conclusions. Thus, we have removed it because a single secondary structure cannot represent the multiple possible structures that may take place and that are considered by the target prediction algorithms. We replaced this figure with just an annotated M63 sequence and its base-pairing targets to better emphasize the extent to which M63 has multiple viable targets within the CcpA regulon. This figure better represents the potential interactions between the sRNA candidate and its potential targets.

Minor comments

L15. Remove the apostrophe in "sRNA's" that indicates possession.

Removed the apostrophe in "sRNA's" to become sRNAs.

L17. Change ", however" to "; however,"

Changed ", however" to "; however,".

L40. I do not understand what the authors are trying to convey when they state "to what extent these candidates may overlap". Overlap how?

We have clarified this section to read:

"Due to the differing sequencing methods, diverse inclusion criteria, S. pneumoniae strain differences, as well as limited follow-up since, it is unclear to what extent candidates identified in different studies have overlapping sequences and functions, and their biological relevance remains ambiguous."

L41. Change "focus" to plural.

Changed "focus" to "focuses".

L44. Sentence needs rewriting. Perhaps, change "reveal more about the" to "have a"

We have revised this sentence, it now reads:

"This study's findings enhance our knowledge of the conservation of small regulatory RNAs across the many Streptococcus pneumoniae strains and highlights a handful that appear likely to have a role in growth or virulence."

L50-52. This is an underestimate of infection and deaths due to pneumococcal disease. The number of deaths is closer to 1-2 million deaths per year. (see PMID 30243584).

We added the appropriate reference to better represent the world-wide burden of S. pneumoniae

"Streptococcus pneumoniae is a Gram-positive bacterium that causes various diseases including pneumonia, meningitis, bacteremia, otitis media and sinusitis. Invasive pneumococcal disease is particularly dangerous in children and the elderly (CDC et al., 2013), and in 2004 was responsible for approximately 4 million illness episodes, 445,000 hospitalizations, and 22,000 deaths in the United States (Huang et al., 2011). In 2016, S. pneumoniae was the leading cause of lower respiratory infection morbidity and mortality globally causing over a million deaths (GBD 2016 Lower Respiratory Infections Collaborators, 2016)."

L56-57. It may be helpful for the authors to provide context to this statement. The reason people

tend to focus on sRNAs encoded in intergenic regions is because they are easier to identify and because an RNA transcribed in an intergenic region that lacks an ORF is deemed more likely to be a functional regulatory RNA than a portion of a protein, tRNA, or rRNA encoded transcript that could potentially be an useless, intermediate in RNA decay.

We have revised this statement to read:

“However, studies seeking to identify sRNAs tend to focus on intergenic regions because RNAs transcribed in regions lacking an ORF are assumed to be more likely to be functional regulators, whereas those identified within ORFs may be an intermediate RNA decay product from a protein-encoding transcript.”

L85-87. The sRNAs do not interact with transposases or the gene involved in carbon metabolism, but rather their respective mRNAs.

Throughout the manuscript we are much more careful with our language around targeting transcripts vs. targeting proteins, including this instance and many others.

L169. Change to "sequences that we examined"

We corrected this typo.

Line 194, Change to past tense, "observed"

Changed “observe” to “observed”.

L211. Table 3 is reference, but it is not clear why. Perhaps the authors mean Table 2?

Mistakenly referenced the next table. Changed “(Table 3)” to “(Table 2)”.

L274. Figure 3 should be Figure 4.

Changed “(Figure 3)” to “(Figure 4a)” to reference the correct figure. Also added a new subfigure, so Figure 4 now has parts a and b.

L286. Change to "; however,"

Changed “, however” to “; however,”.

Re: Spectrum03252-24R1 (Assessing the conservation and targets of putative sRNAs in *Streptococcus pneumoniae*)

Dear Professor Michelle M Meyer,

thank you for submitting your revised manuscript to Microbiology Spectrum.

The two expert reviewers greatly appreciate your revised manuscript and the work you have put into the revision. In addition, based on your revised version both reviewers have a few remaining comments which you will find below. I do hope you can consider these comments prior to the potential acceptance of your nice manuscript.

Revision Guidelines

Sincerely,
Tino Polen
Editor
Microbiology Spectrum

Reviewer #1 (Comments for the Author):

This is an excellent revision of the previous version of this paper. The authors considered all of the reviewers' comments and made many thoughtful changes throughout the manuscript. In many places, the text and figures were completely revised as indicated by the compare file. Overall, the revised manuscript is greatly improved. There are several additional minor changes

that could be made for clarification.

1. Line 90-91. This sentence is a bit awkward. Perhaps, split into two sentences.
2. Line 118 and Table 1. The sRNAs are >97% identical. Which sRNA was used in the subsequent target searches? Or were consensus sequence sRNAs used? This point might be brought out more starting in the section on predicting validated targets (starting line 157).
3. Line 125 and Table S1. It is interesting that the sRNAs are most conserved in the species that are closest evolutionarily (i.e., *S. pneumoniae* with *S. mitis* and *S. oralis*). The other species are more evolutionarily distant. This point might be mentioned.
4. Paragraph starting with line 170. What was the target test set for this analysis? Was it the same six genomes used in the next section? Also, does the 3% non-identity affect the target searches? That is, if the TIGR4 csRNAs are tested against only the TIGR4 genome, do the programs do better than against a set of target genomes? Please clarify.
5. Line 179. The inability to detect validated targets seems to be a "false negative" issue, rather than a "false positive" one. Please consider clarifying.
6. Line 226 and elsewhere. The M45 sRNA is predicted to target the transcript encoding TacF, but not TacF (protein) as written. Please correct this shorthand here, and check elsewhere.
7. Line 319. The authors might mention that other probable targets of M63 are metabolic genes in the CodY and PyrR regulons (Table 3).
8. Figure 1D. Please change "Sinha et al. (D39)" to "Sinha et al. (D39W)". Also, the colors in the table do not match those in the Venn diagram.
9. Figure 2B. Table, right. Please change the pink "Conserved" to "Conserved but repetitive".

Reviewer #2 (Comments for the Author):

In this revised manuscript, Eichelman and Meyer assess the conservation and targets of possible sRNAs in *Streptococcus pneumoniae*. In short, starting with a list of 287 small RNAs that were previously identified empirically via RNA-seq based approaches, Eichelman and Meyer distill this list to a total of 58 probable small regulatory RNAs (sRNAs), i.e., RNAs that regulate gene expression by base-pairing, using various criteria such as if their existence was confirmed by qRT-PCR or northern blotting, they lack certain known RNA elements, and they are conserved among *S. pneumoniae* strains (Figs. 1 and 2). Using sRNA-target prediction programs such as TargetRNA3 and IntaRNA, the authors identify potential targets for these sRNAs, some of which are highlighted in the main body of the manuscript (Figs. 3, 4 and 5).

The significance of this work is based on the fact that there is relatively little known regarding which small RNAs other than the Ccn sRNAs are involved in regulating gene expression through base-pairing interactions and what transcripts are regulated by those sRNAs. Since a matchmaker RNA chaperone has not been identified in *S. pneumoniae*, it will be a laborious endeavor to experimentally identify and validate the set of mRNA targets governed by each sRNA. Given our paucity of knowledge regarding the regulatory function of sRNAs in *S. pneumoniae*, this manuscript makes an important contribution to the field by collating and refining a list of potential pneumococcal sRNAs and identifying potential regulatory functions for a set of them using available bioinformatic approaches. My previous criticisms were positively and adequately addressed by the authors. Some minor typographical errors are highlighted below.

Minor comments:

L70, "Gram" was capitalized in an earlier use, but not here.

L102, it seems like "or" should be placed before "3")"

L176, change to "confirmed that the sequences"

L177, change to "ensure that our inputs"

L201, change to "observed that none"

Reviewer #1 (Comments for the Author):

This is an excellent revision of the previous version of this paper. The authors considered all of the reviewers' comments and made many thoughtful changes throughout the manuscript. In many places, the text and figures were completely revised as indicated by the compare file. Overall, the revised manuscript is greatly improved. There are several additional minor changes that could be made for clarification.

We thank the reviewer for their constructive comments which improved the manuscript in the first round of review.

1. Line 90-91. This sentence is a bit awkward. Perhaps, split into two sentences.

We modified this sentence extensively to clarify the findings of our work. It now reads: *“The predictions suggest 4 putative sRNAs are likely to interact with mRNAs coding for transposases. An additional 3 putative sRNAs have likely targets that include transcripts encoding an H⁺/Cl exchange transporter, RuvB (involved in DNA recombination), and genes involved in carbon metabolism regulation (CcpA regulon), respectively.”*

2. Line 118 and Table 1. The sRNAs are >97% identical. Which sRNA was used in the subsequent target searches? Or were consensus sequence sRNAs used? This point might be brought out more starting in the section on predicting validated targets (starting line 157).

To clarify what sequences were specifically used for target prediction we added clarification to this section.

“For all 58 sRNAs we used IntaRNA, sRNARFTarget, and TargetRNA3 and the in-house sequences generated from the previous sequencing studies (see Methods).”

3. Line 125 and Table S1. It is interesting that the sRNAs are most conserved in the species that are closest evolutionarily (i.e., *S. pneumoniae* with *S. mitis* and *S. oralis*). The other species are more evolutionarily distant. This point might be mentioned.

We added a statement to highlight this finding.

*“Additionally, we see a higher number of sRNAs conserved in the more closely related species, like *S. oralis* and *S. mitis*, than the more distant species like *S. mutans* and *S. pyogenes*.”*

4. Paragraph starting with line 170. What was the target test set for this analysis? Was it the same six genomes used in the next section? Also, does the 3% non-identity affect the target searches? That is, if the TIGR4 csRNAs are tested against only the TIGR4 genome, do the programs do better than against a set of target genomes? Please clarify.

To clarify these issues we now start this paragraph:

*“As a baseline evaluation of the RIP programs, we compared the known and predicted targets of the csRNAs (within six different *S. pneumoniae* strains including D39 and TIGR4 where the*

original sequencing was conducted, see Methods) using IntaRNA, sRNARFTarget, and TargetRNA3 (38)."

We also added the following to our section describing the target prediction of the putative sRNAs.

"The sRNA sequences used for target prediction are derived from the original strain sequenced (TIGR4 and D39W), thus small differences between these genomes and the other S. pneumoniae strains could affect target prediction in other genomes. However, the amount of sequence variation is very small in these loci across the strains, and the consideration of many features in target prediction minimizes concerns that single nucleotide changes will drastically affect the predictions across a set of genomes."

5. Line 179. The inability to detect validated targets seems to be a "false negative" issue, rather than a "false positive" one. Please consider clarifying.

We changed our sentence to read:

"These results support the existing evidence demonstrating that even the best RIP programs suffer from high false positive and negative rates, but can provide informative results."

6. Line 226 and elsewhere. The M45 sRNA is predicted to target the transcript encoding TacF, but not TacF (protein) as written. Please correct this shorthand here, and check elsewhere.

We corrected the shorthand and checked for other similar misattributions.

7. Line 319. The authors might mention that other probable targets of M63 are metabolic genes in the CodY and PyrR regulons (Table 3).

Changed "which has 13 probable targets four of which are in the CcpA regulon" to "13 probable targets including four in the CcpA regulon, and a smattering across the PyrR and CodY regulons."

8. Figure 1D. Please change "Sinha et al. (D39)" to "Sinha et al. (D39W)". Also, the colors in the table do not match those in the Venn diagram.

On Fig. 1D we corrected "D39" to "D39W". We adjusted the opacity of the coloring on the table to match that of the Venn diagram to ensure the colors match more closely.

9. Figure 2B. Table, right. Please change the pink "Conserved" to "Conserved but repetitive".

Changed "Conserved" to "Conserved but repetitive" in Figure 2B.

Reviewer #2 (Comments for the Author):

In this revised manuscript, Eichelman and Meyer assess the conservation and targets of possible

sRNAs in *Streptococcus pneumoniae*. In short, starting with a list of 287 small RNAs that were previously identified empirically via RNA-seq based approaches, Eichelman and Meyer distill this list to a total of 58 probable small regulatory RNAs (sRNAs), i.e., RNAs that regulate gene expression by base-pairing, using various criteria such as if their existence was confirmed by qRT-PCR or northern blotting, they lack certain known RNA elements, and they are conserved among *S. pneumoniae* strains (Figs. 1 and 2). Using sRNA-target prediction programs such as TargetRNA3 and IntaRNA, the authors identify potential targets for these sRNAs, some of which are highlighted in the main body of the manuscript (Figs. 3, 4 and 5).

The significance of this work is based on the fact that there is relatively little known regarding which small RNAs other than the Ccn sRNAs are involved in regulating gene expression through base-pairing interactions and what transcripts are regulated by those sRNAs. Since a matchmaker RNA chaperone has not been identified in *S. pneumoniae*, it will be a laborious endeavor to experimentally identify and validate the set of mRNA targets governed by each sRNA. Given our paucity of knowledge regarding the regulatory function of sRNAs in *S. pneumoniae*, this manuscript makes an important contribution to the field by collating and refining a list of potential pneumococcal sRNAs and identifying potential regulatory functions for a set of them using available bioinformatic approaches. My previous criticisms were positively and adequately addressed by the authors. Some minor typographical errors are highlighted below.

We thank the reviewer for their constructive comments which improved the manuscript in the first round of review and for their appreciation of this work.

Minor comments:

L70, "Gram" was capitalized in an earlier use, but not here.

Changed "gram" to "Gram" and ensured there are no subsequent inconsistencies related to the capitalization.

L102, it seems like "or" should be placed before "3)"

Added "or" before the third item in the list to indicate the "at least" requirement of the criteria.

L176, change to "confirmed that the sequences"

Added "that" after "confirmed".

L177, change to "ensure that our inputs"

Added "that" after "ensure".

L201, change to "observed that none"

Added "that" after "observed".

Re: Spectrum03252-24R2 (Assessing the conservation and targets of putative sRNAs in *Streptococcus pneumoniae*)

Dear Professor Michelle M Meyer,

thank you very much for submitting your revised manuscript!

Your manuscript has been accepted, and I am forwarding it to the ASM production staff for publication. Your paper will first be checked to make sure all elements meet the technical requirements. ASM staff will contact you if anything needs to be revised before copyediting and production can begin. Otherwise, you will be notified when your proofs are ready to be viewed.

Sincerely,
Tino Polen
Editor
Microbiology Spectrum